

# Data driven healthcare insurance system using machine learning and blockchain technologies

Irum Matloob[1], Shoab Khan[2], Bushra Bashir[1], Rukaiya Rukaiya[3], Javed Ali Khan[4] and Hessa Alfraihi[5]

[1] Department of Software Engineering, Fatima Jinnah Women University, Rawalpindi, Pakistan
[2] National University of Science and Technology, Islamabad, Pakistan
[3] Sir Syed University of Engineering and Technology, Islamabad, Pakistan
[4] Department of Computer Science, University of Hertfordshire, Hatfield, Hertfordshire, United Kingdom
[5] Department of Information Systems, College of Computer and Information Sciences, Princess Nourah bint Abdulrahman University, Riyadh, Saudi Arabia

Corresponding author
Irum Matloob,
irum.matloob@fjwu.edu.pk

## ABSTRACT

Healthcare recommendations and insurance have recently been one of the most emerging research areas in health informatics. The fraud in health insurance is becoming increasingly common day by day. To handle healthcare insurance fraud, there is an urgent need for an intelligent system that cannot only identify and monitor doctors' and hospitals' behavior regarding the health services they provide to patients but can also recommend doctors and hospitals to insured employees based on the quality of services they provided previously. This system creates patient and doctor profiles separately, based on their rating. The proposed system combines singular value decomposition (SVD), K-nearest neighbors based collaborative filtering (KNN-based CF), item-based collaborative filtering (Item-based CF), content-based filtering using term frequency-inverse document frequency (TF-IDF), and K-means clustering and probability distributions to recommend doctors and insurance plans. The system measures similarity scores between patients and doctors using cosine similarity, which helps to determine similarity scores and refine the recommendations. This study also uses blockchain technology to automate insurance claims reimbursement. The results are validated using real data from the employees of a local hospital. The system provides recommendations with a root mean square error (RMSE) value of 0.478 and a mean absolute error (MAE) value of 0.0422. The insurance plans developed using the proposed system have reduced the overall expenditure of the local hospital, with a reduction in total expenses. Blockchain technology further helps prevent healthcare fraud. In the proposed system, a healthcare insurance claims reimbursement system is built using smart contract technology on the Ethereum blockchain, ensuring security & transparency and lowering the number of healthcare frauds. The system includes roles for the insurance company, healthcare provider, and patients. It also provides a platform for claim submission, approval, or refusal. In Pakistan, no such system existed before recommending doctors from different hospitals based on their professional conduct or the good health services they provide.

# INTRODUCTION

The insurance industry around the globe faces critical healthcare fraud-related issues (*Kapadiya et al., 2022*). Many research strategies, as discussed in *Stiernstedt & Brooks (2021)*, are being formed to mitigate the risk of healthcare fraud. There is a dire need to provide a solution by which transparency is offered to all elements of the healthcare ecosystem. The key aspects of the healthcare system are patients, healthcare providers, and services. For example, machine learning algorithms can help the healthcare insurance industry analyze vast amounts of data to identify patterns (*Nabrawi & Alanazi, 2023*). Healthcare systems continue to embrace emerging technologies, and we can expect to see significant improvements and cost-effectiveness in the delivery of healthcare services. Most of the medical coverage programs in Pakistan are out of pocket, representing more than 70% of the total healthcare expenditure. This means that most of the coverage of healthcare funding in Pakistan is out of pocket. The achievement of Sustainable Development Goal 3 (SDG 3) depends on the achievement of Universal Health Coverage (UHC), which depends on ensuring access to the necessary healthcare and financial protection. Recommendation systems are developed to help people make decisions, as they use multiple data sources to forecast choices and preferences related to particular products, as discussed in *Tran et al. (2021)*. In addition, it has helped people select insurance products in the insurance sector, as mentioned in *Adomavicius & Tuzhilin (2005)*. Three primary types of recommender systems have been discussed in *Qazi et al. (2020)*: content-based, collaborative, and hybrid filtering.

The two areas that have gained significant interest are (1) the formation of insurance packages based on the requirements of employees and (2) the development of a doctors' recommendation system based on previous practices and patient feedback using machine learning (ML) techniques. The healthcare industry can be revolutionized by having customized insurance package mechanisms and doctor recommendation systems through a web portal, so patients can easily choose their preferred physicians and insurance packages. The system would utilize ML algorithms to analyze patients' clinical and personal data to provide personalized recommendations for healthcare providers based on the patient's specific requirements and preferences. This will also enable patients to make informed decisions about their health and connect with doctors best suited to their needs. For example, patients could input their age, gender, location, and other relevant information into the system. The machine learning algorithms can then generate a list of recommended physicians in their area. This will improve outcomes by ensuring patients are matched with the most suitable doctors. Consequently, this reduces healthcare costs by minimizing the need for ineffective or unnecessary treatments. Ultimately, developing an ML-based doctor recommendation system can improve patients' healthcare quality while simultaneously generating customizable insurance packages.

Since its inception, blockchain technology has been popular in the financial and insurance sectors because it offers a secure, decentralized, and transparent system for storing and transferring data. In insurance claims, blockchain can automate and streamline the claims reimbursement process. Traditional insurance claim processing is intricate, time-consuming, and involves multiple parties, such as healthcare providers, insurance companies, and patients. It can lead to many flaws in the system, *e.g.*, delayed payments or fraudulent activities. Blockchain technology addresses these challenges by creating a decentralized system that stores and transfers insurance claim data. Further, data can not be altered once entered into a blockchain (*Haleem et al., 2021*). Using a system based on blockchain technology, insurance companies can reduce costs, enhance transparency, and improve the accuracy and speed of claims processing.

Current healthcare frameworks face several security vulnerabilities, including data breaches, unauthorized access, and a lack of interoperability among systems. These vulnerabilities can lead to the exposure of sensitive patient information, financial losses, and compromised patient care. Recent studies highlight that traditional healthcare systems often rely on centralized databases, which are more susceptible to attacks (*Kuo, Kim & Ohno-Machado, 2017*; *Agbo, Mahmoud & Eklund, 2019*).

Blockchain technology offers solutions to mitigate these issues through its decentralized, transparent, and tamper-resistant nature:

- Blockchain's cryptographic features protect patient data from unauthorized alterations, ensuring that health records remain accurate and confidential (*Gopalakrishnan et al., 2018*).
- The decentralized nature of blockchain allows all authorized parties to access the same data, reducing discrepancies and building trust among stakeholders (*Gordon & Catalini, 2018*).
- Smart contracts can automate and enforce the terms of insurance policies, reducing administrative overhead and minimizing the risk of fraud (*Hasselgren et al., 2020*).

A blockchain can serve as a secure layer for an insurance reimbursement system's data, a centralized database may still be used for operational efficiency (*Esmaeilzadeh & Mirzaei, 2019*), where non-sensitive information is stored.

## Motivation

Finding the right doctor can be a challenge for patients, especially finding a relevant specialist who meets their needs. The current medical insurance process consists of fixed insurance plans according to the designation of employees and companies providing fixed premium amounts. Furthermore, patients must pay the bills upfront and submit the receipts to the insurance provider for validation, leading to high administration costs, manual verification, and increased fraud risks. This highlights a need for a more efficient and secure insurance claim process.

## Contributions

The proposed system aims to provide

- A user-friendly platform for healthcare providers, insurance companies, and patients.
- Customizable insurance packages and a streamlined insurance claims reimbursement process.
- Recommendation of doctors to improve the accessibility and affordability of healthcare services.

To the best of our knowledge, no such system exists that provides customizable or personalized insurance plans and the recommendation of pertinent doctors in a secure and efficient manner.

## RELATED WORK

A substantial body of work in the recent past, including various projects and evolving technologies, provides valuable information on the background and development trajectory of an health insurance systems that generate personalized health packages and suggests doctors based on patient demands.

For the intended readers, a literature review has been divided into three sub-sections. The first section discusses general healthcare recommendation systems with a particular focus on recommendation systems using machine learning approaches. In the second sub-section, insurance plans and the reimbursement mechanism have been discussed. Then, blockchain technology and its applications in healthcare were discussed.

### Brief overview of existing recommendation system

A recommendation system is an information filtering tool that can predict and suggest products, places, organizations, books, or videos to users based on their preferences and behavior. To date, limited progress has been made in improving healthcare through intelligent systems, and significant gaps remain in developing solutions that can accurately recommend the most suitable doctors based on patient reviews. In *Tabari et al. (2019)*, the authors provided the path for creating a successful decision-making system for effective resource sharing. In *Han et al. (2018)*, a recommendation system is proposed that uses an ML algorithm to predict the user's recommendations based on their previous data. The developers used the consultation histories of numerous physicians to increase patients' trust in their family doctors. In *Waqar et al. (2019)*, it has been argued that improving patient health outcomes and satisfaction with their care is highly dependent on confidence in the doctor-patient relationship. In *Schäfer et al. (2017)*, authors discovered the usefulness of recommendation systems in the healthcare industry that has been published in the past 10 years. A crucial component of the healthcare system was the location of people with the same disease (*Ceyhan, Orhan & Domnori, 2017*). A mechanism was proposed that analyzes a patient's symptoms based on demographic information and recommends appropriate medical tests based on their circumstances using sequential minimal optimization, J48, and random forest algorithms. The authors in *Qian et al. (2019)* have explored the influence of patients' emotions on treatment recommender

systems, incorporating emotional information derived from user evaluations, ratings, and social data. Some other researchers proposed a doctor recommendation system that employed the hybrid algorithm probabilistic matrix factorization integrated with convolutional neural network (PMF-CNN). The PMF-CNN model uses a convolutional neural network to learn the contextual aspects of the revision information and extract a more precise feature representation, thus implementing the revision information modeling. The problem with the PMF-CNN model was that it could only apply historical physician recommendations. The system did not evaluate and recommend new doctors in the network. This restriction led to an additional study criterion that called for adding more elements to the system to address the issue and improve the system's precision and effectiveness. This was addressed in the deep learning framework based on integrated multi-context information proposed in *Yan, Yu & Yan (2020)*.

An overview of several recommender systems approaches and insights into tactics to improve the recommendations' transparency and user understanding have been provided in *Tintarev & Masthoff (2007)*. It discusses strategies like model-agnostic methods, collaborative filtering explanations, content-based explanations, hybrid approaches, and *post-hoc* explanations. In *Mu (2018)*, authors examine how complex patterns and hidden characteristics can be extracted from large datasets using deep learning techniques like convolutional neural networks (CNNs), recurrent neural networks (RNNs), and autoencoders. This improves the accuracy and effectiveness of specific recommendations in various contexts.

The study in *Koren, Bell & Volinsky (2009)*, Investigates user-item interaction matrices and how they can be mathematically broken down into latent elements using matrix factorization techniques to improve recommender systems. This will allow for more precise and efficient recommendations. In *Verbert et al. (2012)*, the authors thoroughly examine the Context-Aware Recommender Systems (CARs). To adjust recommendations depending on contextual aspects, including user preferences, learning objectives, and ambient conditions, authors study various strategies used in CARS, such as collaborative filtering, content-based filtering, and hybrid approaches. To protect user data and improve privacy in recommendation systems for consumer healthcare services, authors in *Katzenbeisser & Petkovic (2008)* explore methods such as secure multiparty computation, differential privacy, and homomorphic encryption. In the recommender system industry, numerous expert systems have recently been developed to address various issues as proposed in *Huang et al. (2012)*; they have implemented an algorithm to improve the performance of the Shanghai Medical League Appointment Platform. This required reserving doctors for patients using a time-sharing system, which overlooked other essential components of managing appointments and allocating time between patients and doctors. The author in *Narducci et al. (2015)* developed a recommender system that identified patients with comparable conditions and suggested renowned doctors based on the semantic relationship between a patient's symptoms and therapy. The system's main problem was the absence of a means to indicate how a patient should rate a particular doctor. In *Salunke, Kasar & Smita (2015)*, the authors developed a doctor profiling tool using natural language processing (NLP) and generated user reviews in their recommender

system. The primary goal was to produce suggestions based on user evaluations; however, it was unclear what characteristics patients used to grade a particular doctor. The proposed idea would have failed to identify any particular solution if no online statistics were available.

On the other hand, the content-based recommendation systems, as discussed in *Horsburgh, Craw & Massie (2015)*, *Ignatov et al. (2016)*, *Pazzani & Billsus (2007)*, operate differently. Every individual and every item has a set of features assigned to it (a profile). The similarity between users and products is gauged using this profile. These characteristics usually stem from an accurate description of the item being recommended; for instance, a movie profile usually includes details about the film's genre (comedies, action, *etc.*), actors, popularity at the box office, release date, and other relevant information (*Shu et al., 2018*). Therefore, an item's features must be characterized to develop a CBF recommendation system. CBF mainly uses similarity metrics' functions, determining how similar or distinct two feature vectors are to directly compare user and item profiles. CBF models utilize the user's historical behaviour to inform their suggestions rather than making a direct user comparison. They extract the desired suggestions from the feature-based representation of the items in the database.

The process of clustering involves arranging a collection of items into groups based on how similar they are to one another compared to other groups (*Na, Xumin & Yong, 2010*). In *Tian et al. (2019)*, the authors discussed hybrid recommender systems and outlined their three primary approaches.

However, there is a cost associated with both diversity and accuracy. That is to say, accuracy must usually be sacrificed to have greater diversity. Still, if user pleasure rises, then this accuracy drop is better. This is a well-known problem with systems recommending things (*Javari & Jalili, 2015*). Researchers studying recommender systems have long concluded that an effective recommender system needs more than just prediction accuracy. For instance, authors in *McNee, Riedl & Konstan (2006)* suggested that criteria other than accuracy should be used when evaluating recommender systems. Novelty and insight are closely associated with diversity since more diverse recommendations will likely introduce users to novel and unexpected items.

In earlier studies, authors have proposed a greedy selection algorithm, for example, in *Zhang (2009)*. In this method, the items are sorted according to their similarity to the target query. Then, the algorithm gradually builds the search set (or a recommendation list) to optimize similarity and diversity.

A performance-based comparison of the recommendation systems is given in Table 1, which helps to identify the best-fit algorithms based on specific use cases, dataset characteristics, and operational constraints.

## Health insurance claims: now & then

An insurance claim is a formal request by the policyholder to the designated insurance company to pay or cover losses.

When an insured event occurs, such as an accident, illness, property damage, or any other covered incident, the policyholder submits a claim to the insurance company to

**Table 1 Performance-based comparison of recommendation systems.**

| Algorithm name | Type | Key features | Performance metrics | Strengths | Weaknesses |
|---|---|---|---|---|---|
| User-based collaborative filtering (*Sarwar, Karypis & Konstan, 2001*) | Collaborative filtering | Uses user-item interaction matrix, finds similar users. | Precision: Medium, Recall: Medium | Simple to implement, interpretable | Struggles with scalability, cold start issue |
| Item-based collaborative (*Sarwar, Karypis & Konstan, 2001*) filtering | Collaborative filtering | Focuses on item similarity, calculates relationships between items. | Precision: Medium, Recall: High | Effective for large user bases | Limited by sparse data |
| Matrix factorization (*Koren, Bell & Volinsky, 2009*) (*e.g.*, SVD) | Collaborative filtering | Latent factors model, reduces dimensions of user-item matrix. | RMSE: Low, Precision: High | High accuracy, handles sparsity well | Requires pre-computation, cold start issue |
| Content-based filtering (*Pazzani, 1999*) | Content-based | Relies on item features (*e.g.*, keywords, metadata). | Precision: Medium, Recall: Medium | No need for user interaction data | Limited diversity, over-specialization |
| Hybrid models (*Burke, 2002*) | Hybrid | Combines collaborative and content-based methods, often includes metadata and contextual info. | RMSE: Very Low, Precision: High | Addresses cold start, improves diversity | Computationally, expensive, complex setup |
| Neural collaborative filtering (NCF) (*He et al., 2017*) | Collaborative filtering (Deep learning) | Uses deep learning architectures like MLPs to model interactions. | RMSE: Low, Precision: Very High | High accuracy, adaptable to complex data | High computational cost, data hungry |
| Autoencoders (*Wang & Chen, 2015*) | Collaborative filtering (Deep learning) | Reconstructs user-item interaction matrix using unsupervised learning. | RMSE: Low, Recall: High | Handles sparse data, flexible architecture | Requires large datasets, cold start issue |
| Graph-based methods (*Ying et al., 2018*) | Graph-based | Models recommendations as a graph (*e.g.*, nodes for users and items, edges for interactions). | Precision: High, Recall: Medium | Excels at capturing relationships | High computational complexity |
| Reinforcement learning (*Ramaswamy & Thrun, 2006*) | Sequential/Contextual | Learns from user interaction feedback over time to optimize long-term rewards. | RMSE: Variable, Precision: High | Personalization, adapts to user behavior | Needs significant interaction data |
| Bandit algorithms (*Li et al., 2010*) | Contextual bandits | Models recommendations as exploration *vs.* exploitation problems. | RMSE: Medium, Precision: Medium | Good for limited user history | Struggles with diverse user groups |

receive reimbursement or coverage according to the terms of their insurance policy. The paperwork-based insurance process is cumbersome and time-consuming. Additionally, it has expensive managerial expenses. This is because the clients must submit their bill receipts to their company before they can be authenticated, approved, and sent to the insurer's bank account.

The research in *van Otterlo & Wiering (2012)* with its insights into automated insurance claim reimbursements along with "Reinforcement Learning" demonstrates how this technology can transform and streamline the claims process, bringing in a new era of efficiency and accuracy in the healthcare insurance sector.

While processing an insurance claim, it has to be ensured that all parties are satisfied and abide by the contract's agreed requirements. Claims payment and arbitration have been extremely challenging, demanding high administrative costs and manual procedures. The manual insurance claim procedure takes significant time and effort to complete.

The claims fraud was a prominent problem that insurance businesses had to deal with as well; the policyholder may give bogus information to get paid for false claims, as discussed in *Al-Karaki et al. (2019)*. The article also claimed to ensure consent management, micropayments, and system interoperability. Despite all the efforts with manual or semi-automated insurance systems, fraud has been a persistent issue. A solution to detect and prevent fraud and risks in the insurance industry was developed using a machine learning approach as in *Guo et al. (2016)*.

## Existing blockchain-based healthcare systems

The present healthcare system faces numerous challenges while sharing and managing the data of patients across several networks and stakeholders. Consequently, it also makes data security and privacy difficult, as discussed in *Alnavar & Babu (2021)*.

Blockchain has become a widely used phenomenon in healthcare in the recent past (*Amanai, 2023*; *MDPI, 2023*; *ArXiv*; *Haleem et al., 2021*; *Tripathi, Ahad & Paiva, 2020*; *Galaba et al., 2023*; *Alnavar & Babu, 2021*). The domain of E-health also uses various blockchain structures to help secure the healthcare system for storing patient information. Currently, most of the health data is stored in centralised databases, which are a lucrative target for cyberattacks. Moreover, patient data persistently remains viewable across the network, making it more vulnerable to attacks. In addition, data integration remains a significant challenge in the healthcare sector. In such an environment, hospitals can address such issues by storing and organizing their data using a decentralized structure like blockchain (*Chen et al., 2021*). The health insurance ecosystem forms a triangle, having service providers, healthcare hospitals, and patients at its three vertices as discussed in *Kuckreja, Nigde & Patil (2021)*. Service providers are the insurance companies that provide insurance to their customers (patients).

In another work (*Zhou, Wang & Sun, 2018*), a blockchain-driven medical insurance storage system, MIStore, was proposed. This approach made it easier for an insurance provider to get a patient's total medical spending records. Moreover, as long as a specific percentage of servers is considered trustworthy, the medical spending data stored in the blockchain is always private to the servers. The blockchain stores all relevant data, and all users may trust the data stored on the blockchain because of its tamper-resistant property. There are a number of consensus algorithms used in blockchain to ensure that all the network stakeholders agree on the validity of transactions. These algorithms also help to secure the network to prevent fraud, making it ideal for fraud detection in healthcare systems. The consensus algorithms used in the blockchain component can be a Proof of Authority (PoA) or a Practical Byzantine Fault Tolerance (PBFT) (*Zhou, Wang & Sun, 2018*), which are suitable for permissioned blockchains typically used in healthcare settings. These algorithms provide faster transaction times and are more energy-efficient than traditional Proof of Work (PoW) systems. Additionally, these algorithms provide robust security while accommodating the performance requirements of healthcare systems (*Mettler, 2016*).

| Table 2 Challenges blockchain can help address. | |
|---|---|
| **Challenges** | **Blockchain solution** |
| Data breaches | Decentralized (*Amanai, 2023*; *ArXiv*) |
| Identity theft | Encrypted self-sovereign identity systems (*Appstek Corporation, 2023*; *MDPI, 2023*) |
| Ransomware attacks | Distributed, redundant systems (*MDPI, 2023*) |
| Fraudulent claims | Transparent transactions (*Appstek Corporation, 2023*; *ArXiv*) |
| Lack of interoperability | Unified data exchange platform (*Amanai, 2023*; *MDPI, 2023*; *ArXiv*) |
| Counterfeit drugs | Transparent drug tracking from production to patient (*MDPI, 2023*) |

Table 2 presents a summary of challenges with various blockchain solutions to address the issues. The research in *Singh et al. (2021)* provides a detailed blockchain architecture for the food industry, assuring security and data integrity. Over the years, there has been a discernible shift toward more technologically enabled, integrated patient-centred care. The emphasis has been on ensuring that healthcare systems are proactive and reactive, utilizing data to anticipate and address patient needs before they develop into serious problems. The proposed system provides a platform for customizable packages and doctor recommendations based on the doctor's good conduct and patient feedback.

## Research gap

The healthcare industry in Pakistan is not fully automated. Although many websites help people get access to book appointments online and access good doctors, they are not centralised. Moreover, no system handles the health insurance-related issues. The proposed system automates insurance industry mechanisms and recommends services, doctors, and hospitals to patients based on their good conduct, performance, and patient feedback. An added capability of the proposed system is to generate insurance plans based on the needs of the employees or clients. If we look closely, none of the studies or projects discussed above integrates all these features into one system.

## MATERIALS AND METHODS

The proposed methodology comprises three main modules. In the first module, collaborative and content-based filtering techniques with clustering generate healthcare recommendations for insured patients. This module is based on doctor and department-related data from four different hospitals in Pakistan. The second module leverages these recommendations to generate customised insurance plans using machine learning algorithms and has been developed and validated using employees' historical data of a local hospital. The third module is a blockchain-based automated claim reimbursement sub-system. It ensures transparent processing of insurance transactions using blockchain technology. Feedback from insured patients further refines the system, while enterprises and insurance companies interact with the framework for plan validation and deployment. All three modules work independently. Figure 1 depicts three modules of

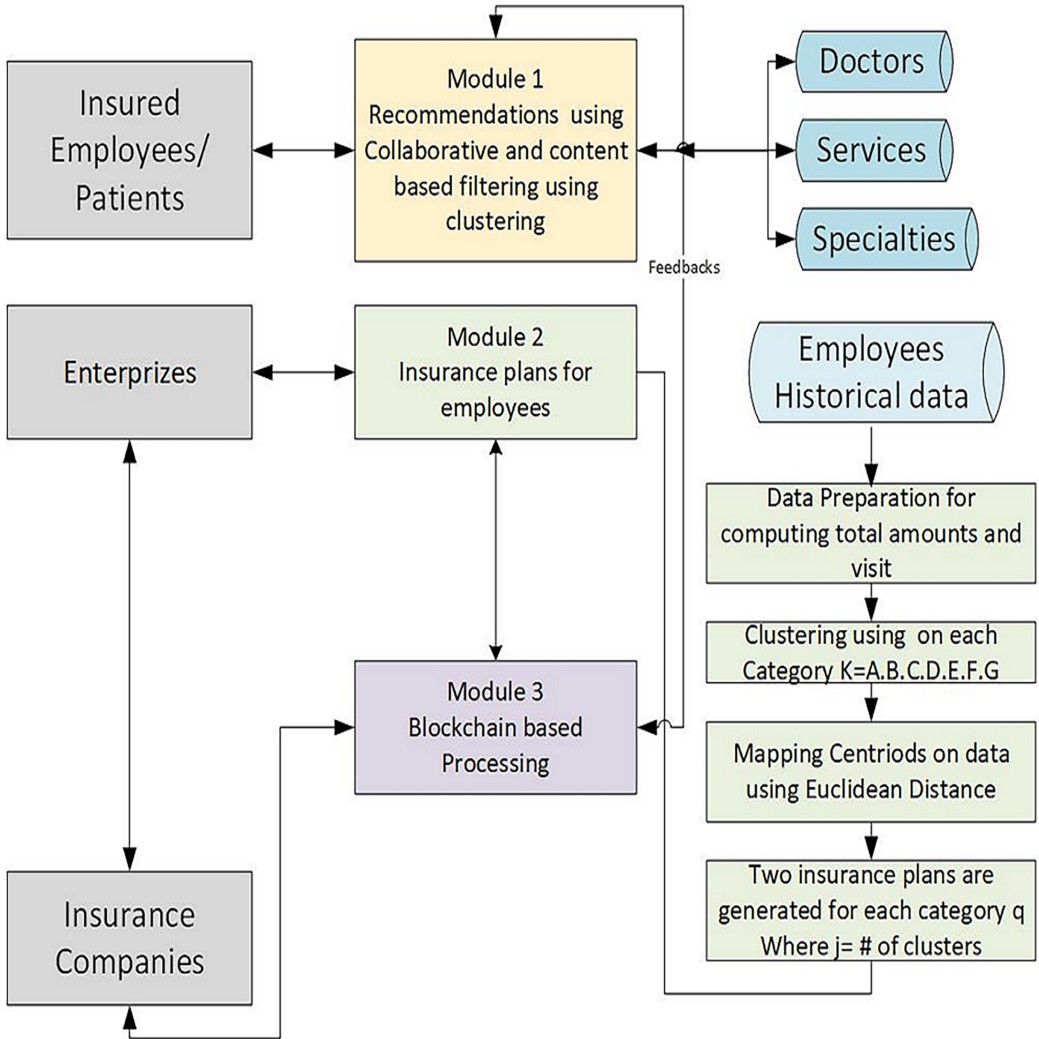

**Figure 1** Overview of the proposed methodology, it is composed of three integrated modules designed to generate personalized insurance plans using historical data and blockchain technology.

the proposed system. The proposed framework emphasises decentralised processing, user feedback integration, and data-driven customisation.

## Module 1: recommender system

For the first module, a hybrid recommendation framework integrates collaborative filtering (CF) techniques, like singular value decomposition (SVD), K-nearest neighbours (KNN), and item-based collaborative filtering (IBCF), with content-based filtering (CBF) to improve the accuracy of doctor recommendation systems. The methodology includes different steps, like data pre-processing, model training, prediction generation, and performance evaluation. In the following, different steps are explained.

## Data processing

This work uses two datasets, *i.e.*,

- The Reviews dataset contains user ratings and textual reviews of doctors.
- The Doctor Information dataset contains metadata about doctors, such as their specialization.

Afterwards, both datasets were combined using Doctor_Name parameter to generate a unified dataset. Then, a user-item matrix is generated, where:

- Users are represented as Rows (Employee_ID)
- Doctors are represented as Columns (Doctor_Name)
- Ratings are represented as Values (Reviews)

Furthermore, the missing values are padded with zeros. For model evaluation, a ratio of 80-20 has been used: training (80%) and testing (20%).

## Collaborative filtering techniques

In this section, we discuss different collaborative filtering techniques in the current context of the proposed recommender system for physicians and services.

### *Singular value decomposition with regularization*

Singular value decomposition or SVD is known as a matrix factorization technique. In the proposed methodology, SVD has been used to decompose the user-item matrix into three components:

- R (user features matrix)
- $\Lambda$ (singular values matrix)
- $Q^T$ (item features matrix)

$$X \approx P\Lambda Q^T$$
$$\hat{X} = P \cdot \Lambda \cdot Q^T$$

where:

- $X$ is the original user-item matrix,
- $P$ is the left singular matrix (user features),
- $\Lambda$ is the diagonal matrix containing singular values,
- $Q^T$ is the right singular matrix (item features),
- $\hat{X}$ is the reconstructed approximation of $X$.

where *Regularization* is applied to reduce the overfitting. Moreover, missing values (unrated items) are predicted by reconstructing the user-item matrix.

### *K-nearest neighbors collaborative filtering*

KNN is known for its memory-based collaborative filtering approach. In the proposed methodology, KNN is used to identify similar doctors based on user preferences. Furthermore, the cosine similarity metric is used to compute distances between different items, and ratings for unrated items are predicted based on the average ratings of the nearest neighbors.

### *Item-based collaborative filtering*

IBCF calculates the cosine similarity between doctors in this study based on user ratings. Doctors with similar ratings are kept together in a group, whereas the similarity matrix is used to predict ratings by combining the ratings of similar items.

## Content-based filtering

Here, with CBF, doctors' recommendations are generated based on textual similarity by analyzing the doctor's names using TF-IDF (Term Frequency-Inverse Document Frequency). TF-IDF transforms doctor names into numerical vectors.

$$\text{TF}-\text{IDF} = \frac{f_{t,d}}{\sum_{t' \in d} f_{t',d}} \times \log\left(\frac{N}{n_t}\right).$$

Moreover, cosine similarity is calculated to identify similar doctors and ratings are predicted based on similar doctors' reviews. Predicted ratings are computed using weighted similarities.

$$S = \cos(\theta) = \frac{A \cdot B}{||A|| \, ||B||}.$$

## Hybrid recommendation model

A hybrid model is implemented to enhance prediction accuracy by combining content-based filtering and item-based collaborative filtering. The model assigns a weight factor (alpha) to balance both approaches.

The hybrid prediction formula is as follows:

$$R_{\text{hybrid}} = \alpha \cdot R_{\text{IBCF}} + (1-\alpha) \cdot R_{\text{CBF}}$$

$$\hat{R} = \frac{\mathbf{S} \cdot \mathbf{U}^T}{\mathbf{S}^T_{\text{sum}} + \varepsilon}$$

where:

- $\hat{R}$ is the predicted ratings matrix,
- $\mathbf{S}$ is the filtered cosine similarity matrix,
- $\mathbf{U}$ is the user-item matrix for common doctors,
- $\mathbf{S}_{\text{sum}}$ is the sum of similarities,
- $\varepsilon$ is a small constant to prevent division by zero.

and

$$\text{RMSE} = \sqrt{\frac{1}{n}\sum_{i=1}^{n}(y_i - \hat{y}_i)^2}$$

$$\text{MAE} = \frac{1}{n}\sum_{i=1}^{n}|y_i - \hat{y}_i|$$

where:

- $y_i$ is the actual value,
- $\hat{y}_i$ is the predicted value,
- $n$ is the total number of samples,
- RMSE penalizes large errors more due to squaring,
- MAE measures the average absolute difference between actual and predicted values.

All the abbreviations used in Algorithms 1 and 2 are listed in Table 3.

We propose two recommender systems, one based on patient reviews and other based on user profiles.

### Generalized/review-based recommender system

In the first recommender system, users are prompted to provide the department names they want to visit. The system then recommends doctors to the user by calculating the similarity scores based on collaborative filtering techniques. The doctors with the highest similarity scores are subsequently recommended to the user. Similarly, enterprises or insurance companies can select attributes based on which they want to design insurance plans for their employees or policyholders.

In Algorithm 1, the input is the hospital department $HD$ of which the user wants the recommendations. $R$ is the empty recommendation list. In our data $F$, the algorithm filters out the $HD$. Now, we only have the data of that particular $HD$. Then, the $AR$ of every $D$ is computed. After that, for every $D_i$, $SS$ is calculated. For $D$ having $SS$ greater than 0 is added to the $R$. And for those $D$ who have $SS$ less than 0 is discarded, and the remaining $D_i$ is checked. After checking all the $D$ of $HD$, a recommendation list is returned.

### Personalized/user-profiling-based recommender system

The second recommender system is based on the concept of user profiling. It asks the user attributes, such as age, gender, and department name, to generate personalised recommendations. The system utilizes the collaborative filtering technique to find similarities between users with similar characteristics and recommends doctors based on the calculated similarity scores. When users interact with the recommender system, they provide their attributes, and the system uses this information to identify other users who share the same characteristics, such as users of similar age, gender, and department preferences. These users form the basis for finding similarity scores. To calculate similarity scores, the system compares the preferences and interactions of the current user with those of other users with similar profiles. It considers factors such as the doctors visited, ratings

---

**Algorithm 1** Generalized/review-based recommender system.

**Input:** $\{HD, UA, UG\}$: Historical data, user age, user gender
**Output:** $R$: Recommended department list
**Initialize:** $R \leftarrow \emptyset$
$D \in HD$ Apply filter function $F(D)$ to preprocess entry
Extract department data $DD$ from $D$
$US_i \in D$ $\delta \leftarrow D[D['\text{Dept\_visited}'] == HD]$
$VC \leftarrow \delta.\text{groupby}('DN')['DN'].count()$
Sort $VC$ in descending order and add $VC[0]$ to $R$
$AgeFilter \leftarrow (D['PA'] \geq UA - 5) \wedge (D['PA'] \leq UA + 5)$
$GenFilter \leftarrow (D['PG'] == UG)$
$@ \leftarrow D[AgeFilter \wedge GenFilter]$
2. **if** $len(@) == 0$ **then** $@ \leftarrow D[GenFilter]$
3.   **if** $len(@) == 0$ **then** $@ \leftarrow D[AgeFilter]$
4.     **if** $len(@) == 0$ **then return** $[\,]$
   $D_i \in \delta$ Calculate similarity score $SS$ for $D_i$
5.       **if** $SS > 0$ Add $D_i$ to $R$
6.       **else** Discard $D_i$ and continue
$R_i \in DD$ Extract $HN, DF, DR, HD$ from $R_i$ return $R$

---

**Algorithm 2** Personalized user profiling-based recommender system.

**Input:** $\{HD, UA, UG\}$: Historical data, user age, user gender
**Output:** $R$: Recommended department list
**Initialize:** $R \leftarrow \emptyset$
$D \in HD$ Apply filter function $F(D)$ to preprocess entry
Extract department data $DD$ from $D$
$US_i \in D\delta \leftarrow D[D['\text{Dept\_visited}'] == HD]$
$VC \leftarrow \delta.\text{groupby}('DN')['DN'].count()$
Sort $VC$ in descending order and add $VC[0]$ to $R$
$AgeFilter \leftarrow (D['PA'] \geq UA - 5) \wedge (D['PA'] \leq UA + 5)$
$GenFilter \leftarrow (D['PG'] == UG)$
$@ \leftarrow D[AgeFilter \wedge GenFilter]$
**if** $len(@) == 0$ **then** $@ \leftarrow D[GenFilter]$
   **if** $len(@) == 0$ **then** $@ \leftarrow D[AgeFilter]$
     **if** $len(@) == 0$ **then return** $[\,]$
$D_i \in \delta$ Calculate similarity score $SS$ for $D_i$
       **if** $SS > 0$ Add $D_i$ to $R$
       **else** Discard $D_i$ and continue
$R_i \in DD$ Extract $HN, DF, DR, HD$ from $R_i$ **return** $R$

---

**Table 3 List of abbreviations used in Algorithm 1 and Algorithm 2.**

| Abbreviation | Meaning | Abbreviation | Meaning |
|---|---|---|---|
| R | Recommendation List | SS | Similarity Scores |
| F | Final Data | HD | Hospital Department |
| AR | Average Ratings | D | Doctor |
| UG | User Gender | UA | User Age |
| PG | Previous User Gender | PA | Previous User Age |
| DD | Department Data | US | User Story |
| * | Ratings Data | @ | Filtered Data |
| HN | Hospital Name | DR | Doctor Review |
| DF | Doctor Fee | VC | Visit Count |
| DN | Doctor Name | $\delta$ | Department Rating |

given, and other relevant information that can capture user preferences. Once the similarity scores are calculated, the system ranks the doctors based on these scores. Doctors frequently visited or highly rated by similar users are given higher scores, indicating a more substantial likelihood of being recommended. The system then generates a list of recommended doctors for the current user. The doctor having the highest number of visits is at the top of the recommended list.

By incorporating user profiling and collaborative filtering, this recommender system provides personalized recommendations that align with the preferences and characteristics of the user. Users with similar profiles are considered to have comparable preferences, making their experiences and recommendations relevant to each other. This approach enhances the accuracy and relevance of the doctor's recommendations, improving the user experience and facilitating informed decision-making. In summary, the user profiling doctor recommender system combines user characteristics, collaborative filtering techniques, and similarity scores to generate personalized doctor recommendations, resulting in a more tailored and effective recommendation system.

The recommended doctor's information includes the doctor's name, hospital name, department, fee, and reviews. This comprehensive set of details allows the user to make informed decisions when selecting a doctor. By considering doctor reviews and user profiling, the system enhances the accuracy and relevance of the recommendations, facilitating an efficient and user-friendly doctor selection process.

Algorithm 2 describes the personalized recommendation system mechanism. The input to the algorithm is the *HD* from which the user wants the recommendations, the *UA*, and the *UG*. *R* is the empty recommendation list. In our data *F*, the algorithm filters out the *HD*. Now, the data of that particular *DD* is left only. After that, the highest visited doctor is found out by filtering the department data $\delta$ and then calculating the number of *VC* of every doctor in *HD*. After that, the highest visited doctor is added to the list *R*. Then, the characteristics of *US_i* are matched with current user characteristics like age and gender. Firstly, the system will try to match the age and gender of *US_i* with given *UA* and *UG*. For

every $US_i$, whose age is $+5$ and$-5$ with given $UA$, and whose gender is the same as $UG$, then the system filters the data (@) according to these $US_i$. But if no $US_i$ have the same age and gender as the current user. Then, the system tries to match $US_i$ age with given $UA$ by following the same process. And still, if no $US_i$ age matches with $UA$, then the system will try to match $US_i$ gender with the given $UG$. After that, for every $D$ in @, $SS$ is calculated. For $SS$ greater than 0, it is added to the $R$ list. And for those $D$ who have $SS$ less than 0 are discarded, and the remaining $D_i$ is checked. After checking all the $D_i$ of $HD$, the recommendation list is returned. Now, for every $D_i$ in the recommendation list, users get their $HN$, $DF$, $DR$ and $HF$ using $DD$.

### Specialties recommendation based on services

The proposed system recommends specialities based on the entered medical service. Only those specialities are recommended whose doctors are of good conduct. As mentioned earlier, historical transactional data from a local hospital of 04 million-plus transactions extended over 5 years has been used for this study. The clustering and content-based filtering have been applied to this transactional data.

The K-means clustering is applied when a service is entered into the system. Based on the entered service, clustering is performed. After this, content-based filtering is applied, and recommendations are generated. Overall, Algorithm 3 depicts the proposed recommender system working.

## Module 2: insurance plans recommender

In this module, recommended insurance plans are considered. Firstly, we create distinct insurance plans for every category. We considered categories of employees used in one of the local hospitals as a case study for generating insurance plans. A complete study of insurance packages and achieved optimization after using those packages is discussed in *Matloob et al. (2021)*.

It is evident from Table 4 that each category type has a specific premium amount. It appears that insurance companies did not take employees' needs into account when creating these packages. Individuals' needs can be assessed based on various factors, including age, gender, marital status, quantity and frequency of visits, *etc*. Considering the previously specified factors, it is evident that each category can be split into two groups according to the needs of any individual. Based on the above-mentioned parameters, the proposed process creates packages by dividing each category into two additional classes.

To create need-based or customised insurance plans, we must analyse transactional data based on categories. One employee category is considered at a time, clustering is applied, and mapping is done, as illustrated in Fig. 2. The x-axis represents the total number of transactions in the selected category, and the y-axis represents the amount that the employees of the chosen category have spent. Groups can be created for each category, as shown in Fig. 2. In each category, the clusters and the distance between each centroid are calculated. All records in the category are under consideration after a centroid for each cluster has been identified. Patients are assigned to each cluster based on the calculated distances between the centroids and all the records in each category. The amounts and

---

**Algorithm 3 Unified recommender system algorithm.**

**Input:** $\{HD, UA, UG, F, N, *, Service_ID\}$
**Initialize:** $R \leftarrow \emptyset$
**for** $D \leftarrow 1N$ in $F$ **do** Filter $HD$ in $F$ Extract department data $DD$
  **for** $US_i \leftarrow 1N$ in $*$ **do** $\delta \leftarrow *[*['Dept_visited'] == HD]$
$VC \leftarrow \delta.$groupby('DN')['DN'].count() Sort $VC$ and add top result $VC[0]$ to $R$
$AgeFilter \leftarrow (*['PA'] \geq UA - 5) \wedge (*['PA'] \leq UA + 5)$ $GenFilter \leftarrow (*['PG'] == UG)$
  $@ \leftarrow *[AgeFilter \wedge GenFilter]$
    **if** len($@$) == 0 **then** $@ \leftarrow *[GenFilter]$
      **if** len($@$) == 0 **then** $@ \leftarrow *[AgeFilter]$
        **if** len($@$) == 0 **then return** [ ]
$D_i \in \delta$ Compute similarity score $SS$
          **if** $SS > 0$ **then** Add $D_i$ to $R$
          **else** Discard $D_i$
$R_i \in DD$ Extract $HN, DF, DR, HD$
**Specialty-Based Recommendation using K-Means:**
Retrieve data for clustering Apply **K-Means** on $Service_ID$ features Assign clusters and
  find top-$N$ similar services
Recommend services from the same cluster
**Collaborative Filtering Methods:**
Compute **SVD** predictions Compute **KNN** similarity Compute **Item-Based CF** using
  cosine similarity
Compute **Content-Based Filtering** using TF-IDF
**Evaluation:**
model $\in$ {SVD, KNN, Item-Based CF, Content-Based CF} Compute RMSE and MAE
  Store results in $R_{metrics}$
**return** Recommended Items $R$

---

**Table 4 Annual premium amount for each category type.**

| Category | Premium (PKR) | Number of employees |
|---|---|---|
| A | 39,036 | 170 |
| B | 17,466.6 | 5 |
| C | 31,922 | 985 |
| D | 13,075.57 | 252 |
| E | 26,309 | 1,991 |
| F | 8,862 | 1,619 |
| G | 6,8811 | 120 |

visits are maintained at the centroid's values for every record. For each centroid, the same steps are repeated. Generating packages requires a sum of the total amounts of all members in one group. Finally, we compute distinct insurance plans for every category.

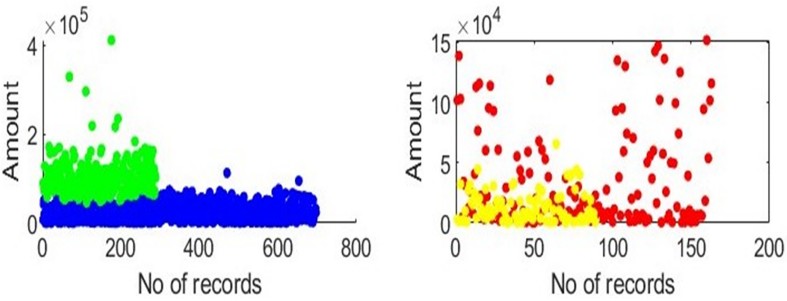

**Figure 2 Clusters depiction: clusters depict that in each category, there are two groups of employees based on amount, gender, age, and visits.**

The proposed system can help enterprises or insurance companies select relevant attributes for employees or policyholders based on which they need insurance plans. Previously, insurance plans were fixed with premium amounts as well. However, now, using the proposed methodology, they can generate customizable insurance plans with customizable premium amounts.

Algorithm 4 describes how the need-based insurance plan generation mechanism can be implemented. The number of clusters, max_iterations, and number of categories $\phi$ are considered as input attributes. The K-means algorithm is first run for a single category. Then the second *for* loop is on several clusters $y$, where $C_y$ indicates the centroid of cluster $y$. $C_y$ is mapped to particular category data. In this case, the term "mapping" has a different meaning: it refers to calculating the distance $C_y$ from each record in the considered category. The fourth iteration of the for loop counts the records in a given category. The distance of a record from the centroid is indicated by $D_r$, where $r$ is the category subscript. One computes the mean of all the $D_r$. The $G_y$ is increased by all the records whose $D_r$ is less than the mean. The remainder are all thrown away. Line number 18 shows that $Am_r$ is the amount of the considered record, and $Am_y$ is the amount of the centroid. $Am_y$ is used in place of $Am_r$. Lastly, the total of all the amounts $Am_y$ in a given group is calculated. To create packages, the total amount of each member in a group must be added. The total sum is calculated for each group $\omega_i$ inside each category $r$. The sum of $\beta_j$ for every category is calculated.

## Module 3: automated insurance claims reimbursement

The proposed effort involves developing a system for processing healthcare insurance claims reimbursements using blockchain technology that eliminates the fraud and risk in the manual process of insurance claims reimbursements. Specifically, the smart contract technology is used for the healthcare reimbursement process. It allows us to create a decentralized system that could manage claims securely and transparently without the involvement of a third entity. To deploy the smart contract on a blockchain, we have used the MetaMask extension, which allows interaction with the blockchain directly from our

---

**Algorithm 4 Customizable insurance plan generation.**

**Input:** $\phi = \{\phi_1, \phi_2, \ldots, \phi_7\}$, number of clusters $C$, max_iterations, and *attributes* $= \{age_r, gender_r, status_r, Amount_r, visits_r\}$

**Output:** $\alpha$

**for** $r \leftarrow 1$ 7 **do** Call **K-means** on $Q_r$

cluster $Q$

    **for** $y \leftarrow 1$ $C$ **do** Find instances closest to centroid $C_y$

$C_y = \{agec_y, genderc_y, statusc_y, Amountc_y, Visitc_y\}$

$C_y$ Map $C_y$ to $Q$

        **for** $r \leftarrow 1$ $|Q|$ **do**

$D_y = \sqrt{(ac_y - a_r)^2 + (gc_y - g_r)^2 + (sc_y - s_r)^2 + (Amc_y - Am_r)^2 + (Vc_y - V_r)^2}$

Calculate Mean $D_r$ Set Mean as $M_y$

*[l] Match grouping of members to centroid

           **if** $D_r \leq M_y$ **then** Add $D_r \rightarrow G_y$

           **else** Discard

Replace $Am_r$ with $Amc_y$ in all members in $C_y$ $Sum_A mount+ = Amc_y$

Generate insurance packages for each category $r$

---

browser. An enriched front-end test-bed has also been developed to support and validate the recommender and reimbursement system. It leverages healthcare providers to submit claims on behalf of a patient quickly. To integrate the smart contract and front-end, we have used the Web3 library, which provides a set of tools for interacting with the Ethereum-based blockchain. It creates a seamless user experience that abstracts away the complexity of the underlying blockchain.

### Solidity smart contract

The healthcare reimbursement system is implemented using the Solidity programming language in Solidity version 0.8.0. The smart contract has been put into operation on the Ethereum network using the Remix IDE and the MetaMask browser extension.

### Contract compilation

The contract includes three main roles: the patient, the healthcare provider, and the insurance company. These roles are implemented as separate addresses with specific permissions and functionalities.

- The insurance company is the entity responsible for verifying patients and approving claims.
- The healthcare provider is the entity that provides healthcare services to the patient and submits reimbursement claims to the insurance company on behalf of the patient.
- The patient is the entity that receives healthcare services.

To implement the above-mentioned entities, we have used the following solidity concepts:

**Structs:** We use two structs: Patient and claim. A boolean value in the patient struct shows whether or not the insurance provider has validated the patient. The claim ID, patient's address, healthcare provider's address, claim amount, and approval/payment status are all included in the claim structs.

**Mappings:** Again, we have used two mappings: patients and claims. The patients mapping maps patient addresses to Patient structs, while the claims mapping maps claim IDs to Claim structs.

**Events:** We use events to emit claim submission, approval, and denial. These events help to provide transparency and enable stakeholders to track the progress of the reimbursement process.

**Modifiers:** We used the "require" keyword with various conditions to ensure that only authorized parties can perform specific actions. For example, only the healthcare provider can submit a claim; only the insurance company can verify patients and approve/deny claims, and only claims that have not been paid can be approved or denied.
The contract has the following main functions:

1. The *verifyPatient* function is used by the insurance company to verify the patient's identity. Only the insurance company is authorized to call this function.

2. The *submitClaim* function is used by the healthcare provider to submit a reimbursement claim on behalf of a patient. Only healthcare providers are authorized to call this function. The function checks whether the patient has been verified by the insurance company and stores the reimbursement claim in the claims mapping.

3. The *approveClaim* function is used by the insurance company to approve a reimbursement claim. Only the insurance company is authorized to call this function, which takes a single parameter: the claim ID. If the reimbursement amount is more than or equal to 1, the function emits the ClaimApproved event to indicate that the claim has been approved.

4. The *denyClaim* function is also used by the insurance company to deny a reimbursement claim. The function takes the claim ID as an input parameter, and only the insurance company is authorised to call this function.

### Contract implementation

Algorithm 5 is used for smart contract implementation and all the abbreviations used in the algorithm are listed in Table 5.

The algorithm describes the Healthcare Reimbursement *"HR"* contract, which governs the processing of claims *(C)* between an *IC* and *HP*. The contract permits patient verification *(vP)*, claim filing *(sC)*, claim approval *(aC)*, and claim denial *(dC)*, and emits related events *(E)* for each action. It includes *M* for storing and retrieving *P* and *C*

**Algorithm 5** Automated insurance claims reimbursements.

**Input:** $\{\_C\_ID, \_\alpha, \_\beta\}$
**if** *msg.sender* $==$ *IC* **then** $P[_\alpha].isV = true$
**else** Throw *E*
   **if** *msg.sender* $==$ *HP* **then**
      **if** $P[_\alpha].isV$ **then** Create new *C* and Store in *M*
Emit *CS*
       **else** Throw *E*
       **else** Throw *E*
       **if** *msg.sender* $==$ *IC* **then**
         **if** $!C[\_C\_ID].isA$ **then** $C[\_C\_ID].isA = true$
           **if** $C[\_C\_ID]. >= 1 \$$ **then** Emit *CA*
           **else** Throw *E*
           Throw *E*
       **if** *msg.sender* $==$ *IC* **then**
         **if** $!C[\_C\_ID].isA$ **then** $C[\_C\_ID].isA = false$
Emit *CD*
         **else** Throw *E*
       **else** Throw *E*

**Table 5 List of abbreviations for Algorithm 4.**

| Abbreviation | Meaning | Abbreviation | Meaning |
|---|---|---|---|
| HR | Healthcare Reimbursement Contract | IC | Insurance Company |
| HP | Healthcare PRovider | P | Patient |
| C | Claim | $\alpha$ | Patient Address |
| C_ID | ClaimID | $\beta$ | Claim Amount |
| isV | isVerified | isA | isApproved |
| _IC | _Insurance COmpany | _HP | _Healthcare Provider |
| CS | Claim Submitted EvEnt | CA | Claim Approved Event |
| CD | Claim Denied Event | _C_ID | _ClaimID |
| _$\alpha$ | _PatientAddress | _$\beta$ | _ClaimAmount |
| £ | Ether | $ | Amount |
| vP | Verify Patient | sC | Submit Claim |
| aC | Approve Claim | dC | Deny Claim |
| Emit | Emit | E | Error |

information. Only the *IC* can *vP*, *aC* or *dC*, and only the *HP* can *sC*. _$\alpha$ represents the patient address for verification which is used as input for *vP* function. *C_ID*, _$\alpha$, _$\beta$ are the parameters that are used as input for *sC* function. _*C_ID* is used as input for *aC* and *dC* function. An *event* is emitted to indicate approval when *C* is accepted and fulfils a certain threshold. Similar to this, an *event* is emitted when the *C* is denied.

### Contract deployment

The Healthcare Reimbursement smart contract is deployed on the blockchain using the Remix IDE and the Metamask injected provider environment. The Metamask environment provides an intuitive interface for interacting with the blockchain and enables the creation of accounts for the patient, healthcare provider, and insurance company. The Metamask browser extension is installed, and test ethers are used to perform transactions. Test ethers are a type of cryptocurrency used in testing blockchain applications and are not real money. Once the contract is deployed, we set the addresses of the healthcare provider and the insurance company by calling the constructor.

By invoking the *verifyPatient* function and providing the patient's address, the insurance company can afterwards validate a patient. This function sets the "isVerified" flag for the patient to true. The healthcare provider can submit a claim by calling the *submitClaim* function and passing in the claim ID, patient address, and claim amount. This function creates a new claim and sets its initial status to false for approval and payment. The insurance company can then approve or deny a claim by calling the *approveClaim* or *denyClaim* function and passing in the claim ID. If the claim is approved, an event is emitted.

There is no consent required for the data as we have de-identified it.

The architecture chosen for the proposed system is multi-tier, which consists of multiple tiers (layers), including the presentation layer, application layer, data layer, and blockchain layer, as shown in Fig. 3. The details of the different tiers are as follows:

**Presentation layer** includes the user authentication process, allowing secure access to the system for patients, healthcare providers and insurance companies. The insurance provider can use the insurance claims verification/approval component of the user interface to check filed claims and approve or refuse them. Users can adjust their preferences, such as choosing preferred doctors based on geography, age, gender, and ratings. Based on the patients' selected departments and ratings, the recommendation component offers generalized recommendations for them. By taking into account the patient's age and gender, and previous user interactions, customised recommendations can also be generated for patients.

**Application layer**

- **Insurance claims processing**
  This component manages the processing of insurance claims, including patient verification, claim approval and denial.
- **Recommendation engine**
  This component generates doctor suggestions for patients based on their preferences and other characteristics using machine learning algorithms, doctors' and past users' data.
- **Similarity calculation**
  Based on characteristics such as age and gender, this component determines how similar patients and doctors are.

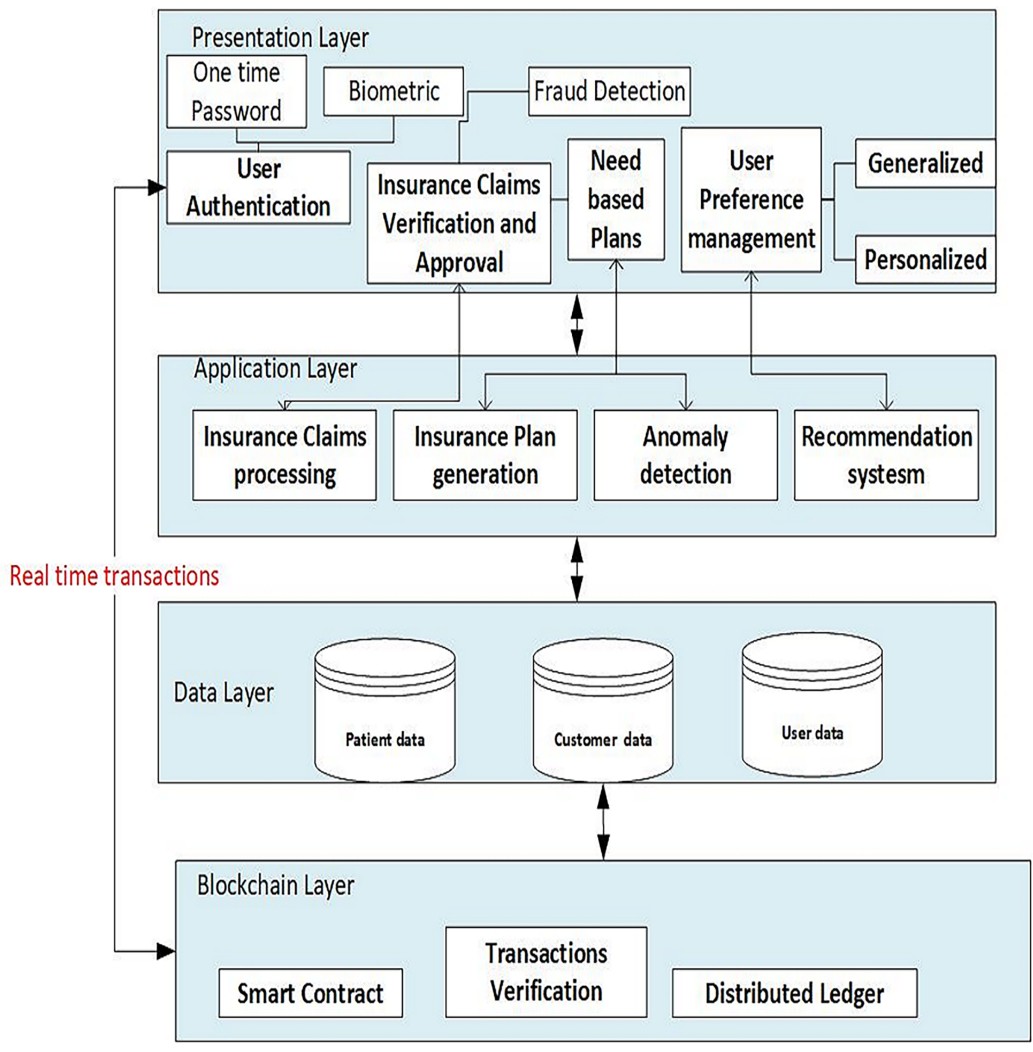

**Figure 3 Architecture of the proposed framework.** The system is structured into four primary layers: the presentation layer, application layer, data layer, and blockchain layer.

- **Collaborative content based filtering**
  This component uses collaborative and content-based filtering techniques to find similar patients, services and evaluate trends of doctor recommendations.

**Data layer**

- **Patient data/Employee data**
  Personal information such as age and gender, medical history including department visited, preferences, ratings of doctors, and records of previous interactions are all stored in this layer together with other patient-related data.
- **Doctor data**
  The specialities, qualifications, names, healthcare processing fees and other pertinent information about doctors are all stored in this layer.

**Blockchain layer**

- **Smart contracts**
  This component uses blockchain-based smart contracts to describe and carry out the business logic of the insurance claim reimbursement process.
- **Transaction validation**
  This component ensures that insurance claim-related transactions adhere to established rules and regulations by verifying their integrity and authenticity.
- **Distributed ledger**
  Transparency and security are provided by this component, which keeps a decentralized and unchangeable record of all insurance claims transactions.

### Overview of components/modules

For improved readability, we present an overview of the different modules of the proposed system in this section.

1. **User or patient module**

This module lets users input their data and preferences, such as age, gender, and hospital department selection. The module also presents the recommended doctors and additional information such as their profiles, hospitals, fees, ratings, and departments.

2. **Healthcare provider module**

This module enables hospitals to access the decentralized server to submit claims on behalf of patients. It also allows healthcare providers to select attributes for generating the insurance plans.

3. **Insurance company module**

This module enables insurance companies to access the decentralized server to verify hospital claims and process reimbursement requests. The module allows insurance companies to access user or patient claims and history data. It also enables insurance companies to select suitable attributes for generating customized insurance plans for policyholders.

Blockchain is specifically used to store historical visit records in Module 1 and to store the recommendations generated in Module 2. In healthcare systems, we need this to guarantee data integrity and traceability, essential for gaining confidence. Additionally, we have implemented a feedback mechanism that feeds Modules 1 and 2 with blockchain-verified data. Over time, this enhances the precision of data analysis and recommendations by enabling real-time updates and ongoing learning.

The system's three modules work together to streamline the reimbursement process and provide personalized and relevant doctor and insurance plan recommendations to users, including patients, healthcare providers, and insurance plans.

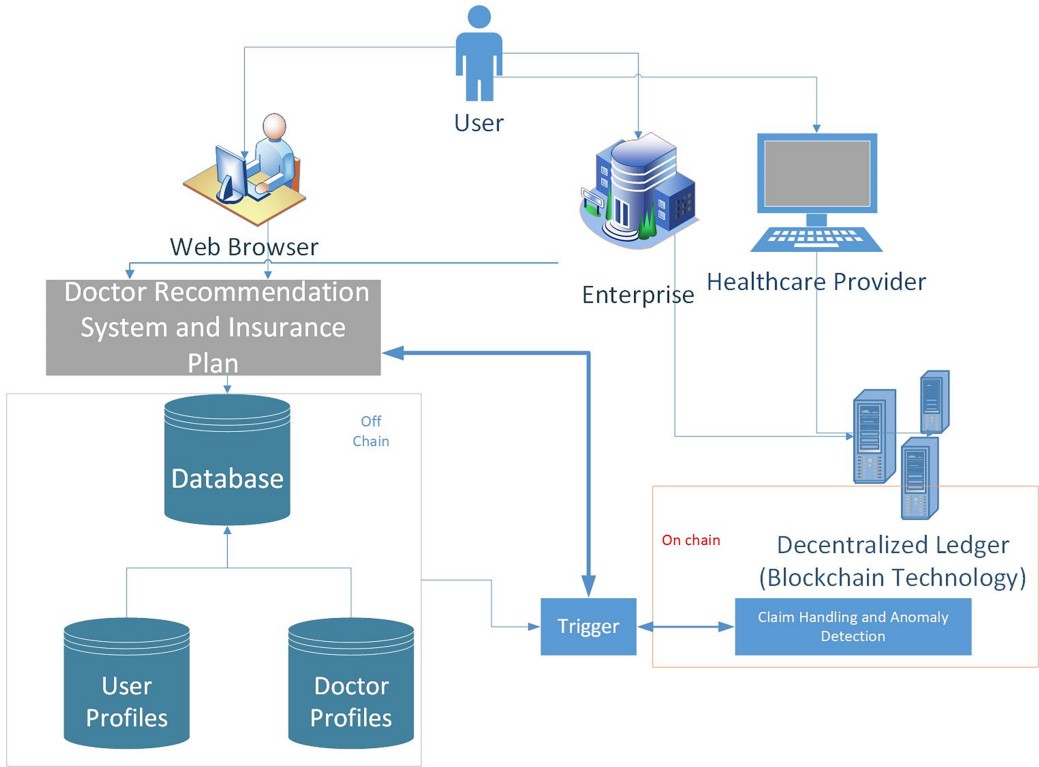

**Figure 4** A hybrid real-time model which handles real-time claims in two ways: using event triggers to stream transactions directly to the blockchain after real-time off-chain validation, and employs smart contracts that automatically enforce rules as transactions are processed.

However, as depicted in Fig. 4, sending all raw transactions and insurance claims to the blockchain can affect the system's performance. Therefore, we have used the concept of on-chain and off-chain data. Raw transactions and claims can be stored in local repositories or data servers referred to as off-chain, whereas the metadata/hash of the data is on-chain. There is an interface layer that connects these two types of data. Furthermore, Machine learning based analytics can be made on off-chain and on-chain data

1. **Interaction of the users:**
   Users connect to the system using web browsers installed on mobile, laptop, or desktop computers.
2. **Recommendation system:** The *Doctor Recommendation System* manages the user preferences and provides certain recommendations.

The module's functionality includes:

- Gathering user preferences and data,
- Analyzing user information and generating doctors' recommendations,
- Providing generalized or personalized suggestions for doctor recommendations.

3. **Insurance company:** The *Insurance Company* manages the insurance-related procedures and components that evaluate and approve claims.
The module's functionality includes:

   ○ Processing claims for insurance.
   ○ Verifying claim details and insurance coverage.
   ○ Evaluating the eligibility of a claim.
   ○ Calculating the amount of reimbursement.
   ○ Generating smart contracts for claims.

4. **Healthcare provider:** *Healthcare Providers* access the system to submit claims and provide medical care.
The module's functionality includes:

   ○ Healthcare providers submitting claims.
   ○ Interacting with the insurance company for claim processing and reimbursement.

5. **Decentralized ledger:** The *Blockchain Infrastructure* represents the decentralized ledger and blockchain layer of the system. The module's functionality includes:

   ○ Storing the distributed ledger, smart contracts, and transaction data is all part of the system's workflow.

6. **Database:** Data on patients and doctors is stored and managed in the database. The system's workflow, as depicted in Fig. 4 it includes:

   ○ Storing and managing patient data, such as contact information, medical history, preferences, and correspondences from the past.
   ○ Archiving and managing medical data, such as specialities, locations, rankings, and other relevant information.

The workflow diagram focuses on the different components in the system. In a nutshell, Users access the *Doctor Recommendation System via* their web browsers, and it processes user preferences and data to produce doctor suggestions. The *Insurance Company* manages insurance claims and collaborates with *Healthcare Providers* to process and reimburse claims. The *Database* stores and manages patient and doctor data for quick retrieval and storage, while the *Blockchain Infrastructure* enables safe and transparent transaction management.

## RESULTS AND DISCUSSION

### Data acquisition

As mentioned earlier, the datasets used in this recommender system are of two types. One dataset contains the information of doctors, *i.e.*, hospital name, department, fee, and qualifications. The doctors' data from four hospitals, including private and public sector hospitals, were collected. The other dataset contains information on patients who have

Table 6 List of recommended doctors.

| User_ID | Age | Gender | Department | Doctor | Hospital |
|---------|-----|--------|------------|--------|----------|
| E_67 | 34 | Female | Dermatology | H1D060 | H3 |
| E_84 | 39 | Female | Dermatology | H1D060 | H3 |
| E_16 | 45 | Female | Dermatology | H1D060 | H1 |
| E_16 | 46 | Female | Dermatology | H1D062 | H1 |
| E_49 | 34 | Female | Dermatology | H1D062 | H1 |
| E_9 | 35 | Female | Dermatology | H3D012 | H3 |
| E_67 | 35 | Female | Dermatology | H4D038 | H4 |
| E_77 | 33 | Female | Dermatology | H1D063 | H1 |

visited doctors of these four hospitals and given their reviews. This dataset contains the user ID, user gender, user age, department visited, hospital name, doctor name, and fee. The data has been preprocessed to remove any missing values or outliers.

### User profiles

The user profile data includes employee age, department visited, department ID, hospital name, doctor name, reviews, and fee. The age distribution of the user-profiling dataset shows that the age group of 35 years is observed to dominate over the others which is depicted in Fig. A1 in Appendix. We also performed the gender distribution of the user-profiling dataset, which concludes that average ratings of hospitals given by user profiles are shown in Fig. A2 in Appendix. The graph in A2 shows the hospital rating spread. We can see that the ratings given to the H2 hospital's doctors lie mainly between 3 and 4, the rating given to most of the H1 doctors is 4, the rating given to most of the H3 doctors is between 3 and 4, and the rating given to most of the H4 doctors is 4.

## Recommendation system

The proposed recommender system is intended to provide generalized and personalized recommendations based on the user's input, department, age, sex, *etc.*

### Home

We have two options for getting recommendations: generalized or personalized. For generalized recommendations, we need to select the hospital department. To find the dermatologist with the highest number of reviews, we chose the dermatology department and clicked the "submit" button. Then, we sorted the dermatologist's profile from the highest reviews to the lowest reviews. Our recommender system recommends that doctor IDs: H1D061, H1D063, H3D012, H3D007, H4D037, H4D038, H4D040, and H1D062 are the highest reviewed doctor. Their rating is 4.0 and an appointment fee of 3,300 or 1,000, as illustrated in Table 6.

If we want to find a dermatologist, but with personalized recommendations. Then, after selecting 'Personalized Recommendations', we will enter age and gender.

After giving input, we get recommendations.

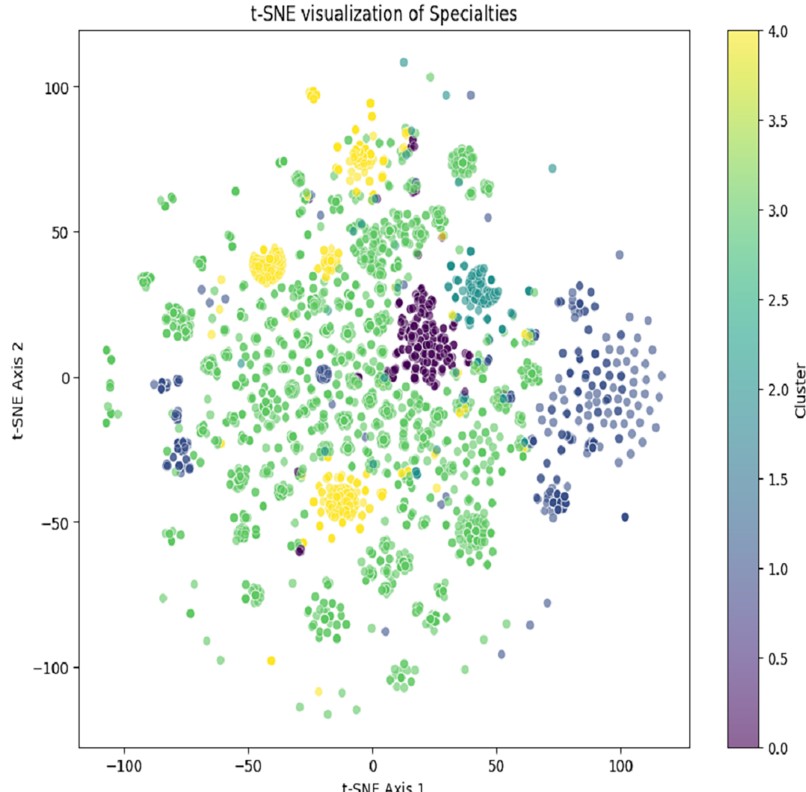

**Figure 5** Generated clusters after using K-means clustering to generate content recommendations.

We have given the input of age '34' and gender 'female'. So, the algorithm can compute the similarity with the previous user profiles and return those doctors visited by the previous users, having age groups between 34 + 5 and 34 − 5 and gender female. Also, in the list of recommended doctor profiles, the top doctor, H3D008, is the most visited dermatologist.

The previous user profiles who visited the above doctors are given below:

All the user profiles' age lies between 34 + 5 and have the same gender 'female' as the current user. The recommendations of the specialties based on the services are generated by using K-means and content-based filtering. The K-means clustering result is depicted in Fig. 5. Selected specialty based on the number of patients using the same specialty for the service.

Table 7 depicts generated recommendations based on service usage for the Selected Specialty.

A greater number of transactions for this C-Reactive Proteins Service are in the Paediatrician and Emergency departments, as depicted in Fig. 6. Similarly, all specialities are analyzed for each service. The subset of data is used for depicting in Fig. 7.

The heatmaps use three distinct similarity metrics—Cosine, Pearson, and Jaccard—to show doctor-to-doctor similarity. Self-resemblance is represented by the diagonal (value = 1), whereas different levels of similarity are shown by off-diagonal values. Distinct

**Table 7 Selected specialty is neurosurgery and recommended and selected specialties based on service usage.**

| Recommended specialties |
| --- |
| Emergency |
| Pulmonologist |
| Neurosurgery |
| Orthopedic |
| OB/Gyne |
| ENT specialist |
| Medical specialist |
| Cardiologist |
| Pain management |
| Dermatologist |
| Rheumatologist |
| General surgeon |
| Physical Med & Rehabilitation |
| Anesthesiologist |
| Opthalmologist |
| Endocrinologist |
| Pediatrician |
| NeUrologist |
| NeUrologist 524 |
| Nutritionist |
| Emergency (501) |
| Infectious Diseases |
| Surgery—Plastic |
| Psychiatrist |
| Urologist |
| Anesthesiologist 519 |
| Pediatric surgeon |
| Surgery—Cardiac |
| Pediatric cardiologist |
| Gastroenterogist |
| Blood bank staff |

similarity clusters show how well the feature representation works by implying significant ties between doctors. Furthermore, the Cosine similarity heatmap depicts

- The values on the heatmap range between −1 and 1. Where a value of 1 means the doctors are identical (same reviews, same department, *etc.*)
- A value of 0 means no similarity.
- A value of −1 means completely opposite, *i.e.*, negative correlation.

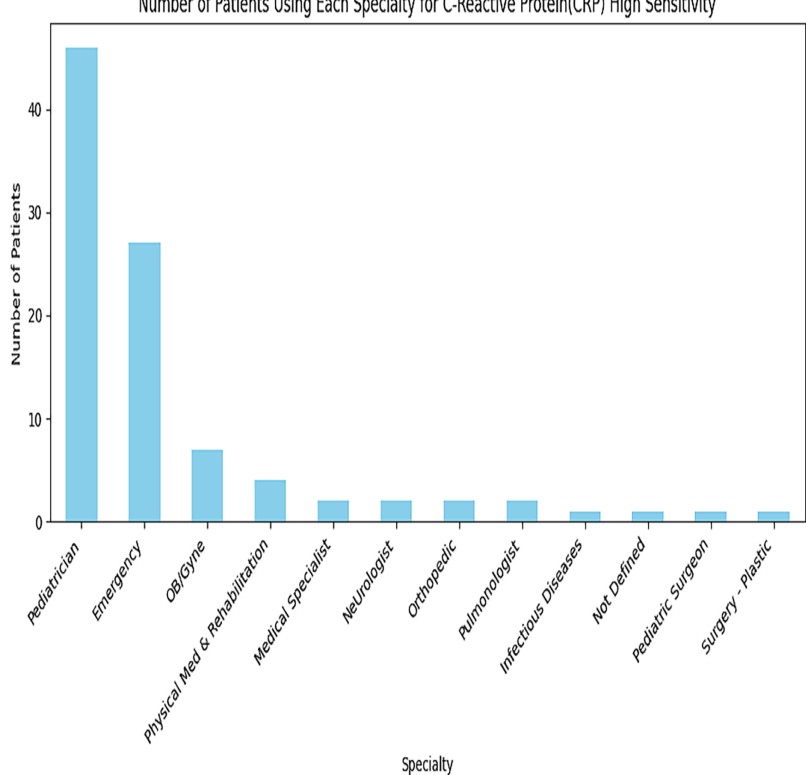

**Figure 6 Recommendations for C-reactive proteins service.**

The heatmap also uses a color gradient to indicate such values, where high similarity is designated with dark colors (*e.g.*, dark blue or dark red) and lower similarity is shown with light colors (*e.g.*, light yellow or white).

Pearson correlation heatmap shows how strongly different doctors' reviews correlate. Where,

- 1 shows a positive correlation
- 0 shows that there is no correlation
- −1 shows a perfect negative correlation.

The Jaccard similarity heatmap shows how similar the doctors are in terms of common attributes. Where,

- A value of 1 means that the doctors have similar sets of features (*e.g.*, the same reviews or department).
- A value of 0 means there are no common features. Heatmap Example Interpretation:
- If the heatmap has dark blue squares along the diagonal, it means that each doctor is highly similar to themselves (self-similarity, which is always 1).
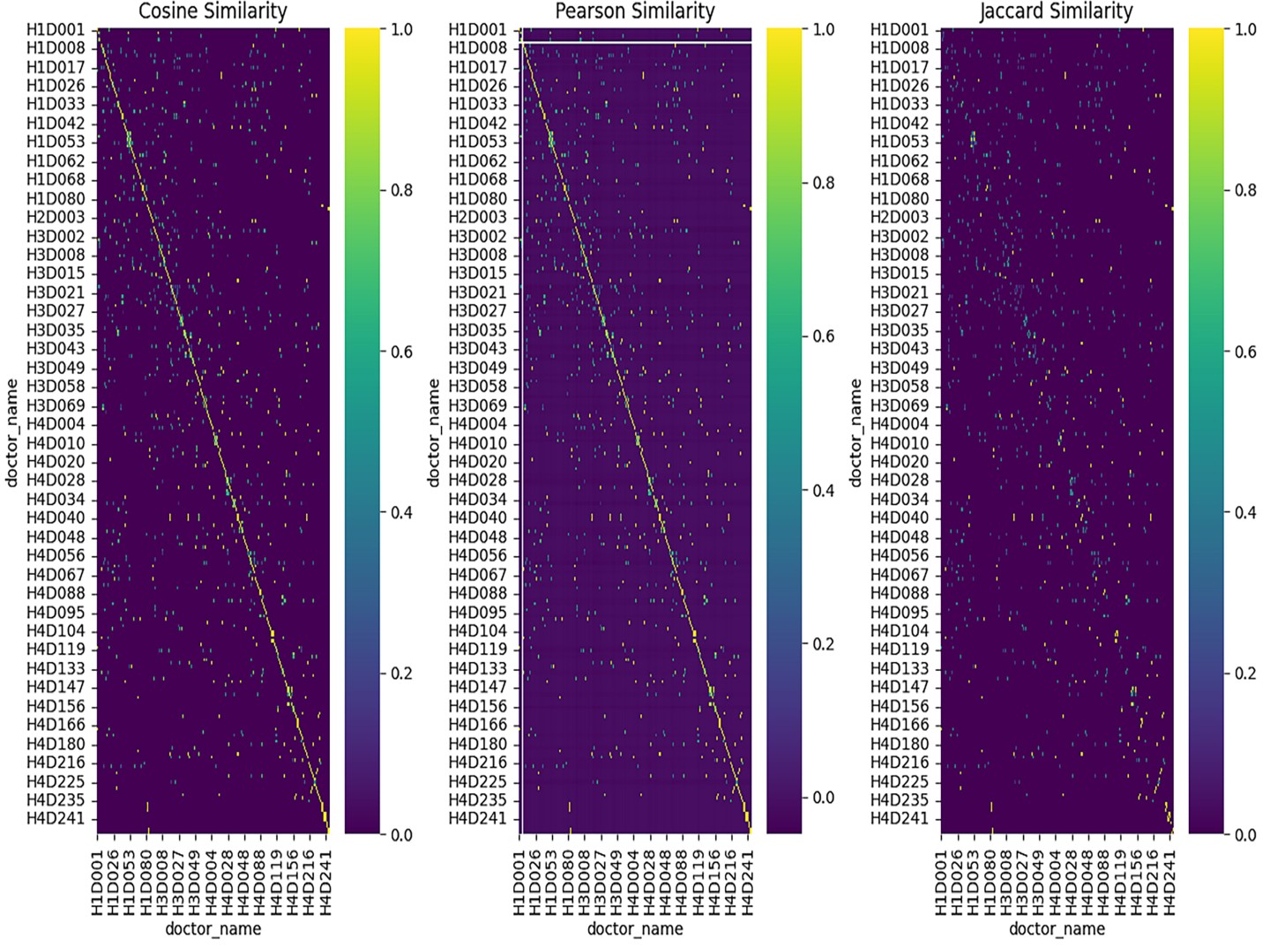

**Figure 7  Heatmap for similarity methods.**

- Whereas, off-diagonal dark squares represent doctors who are very similar to each other based on the chosen similarity metric. For instance, doctors who share similar reviews or departments are put under this category.
- The lighter squares in the heatmap designate the pairs of doctors with lower similarity.

This plot depicts a grouped bar chart to compare the performance (in terms of MSE and MAE) of three similarity methods (Cosine, Jaccard, and Pearson). MAE values are displayed in orange, whereas MSE values are displayed in blue as depicted in Fig. 8.

**Hospitals:** The RMSE of all applied models is depicted in Fig. 9.

The comparison of three models based on mean absolute error is depicted in Fig. 10.

One of the main aspects of the proposed system is the customizeable insurance plan generation. For this purpose, as we have mentioned earlier, we have used local hospital

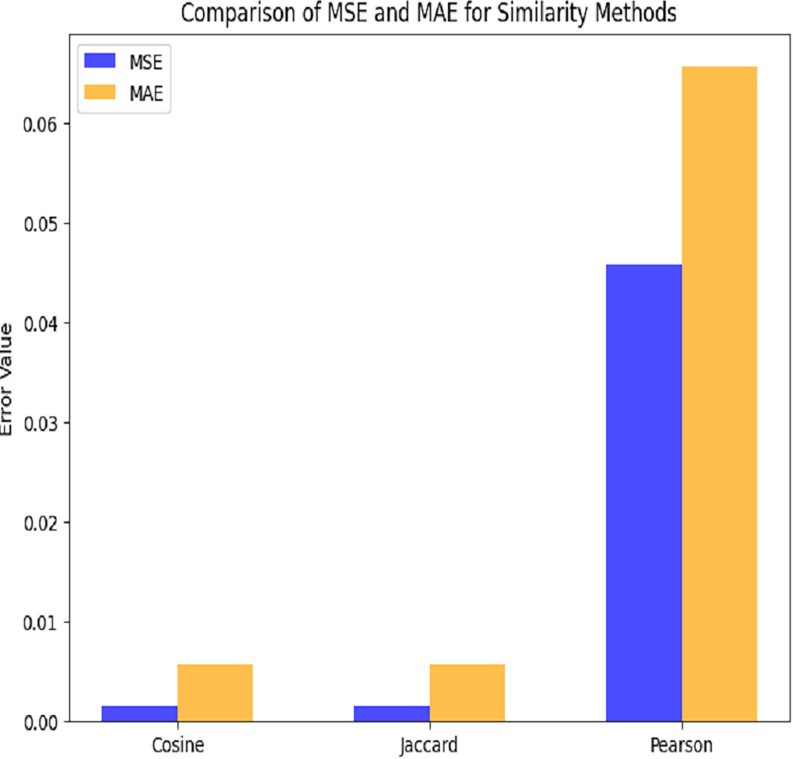

**Figure 8** **Comparison among similarity functions.**

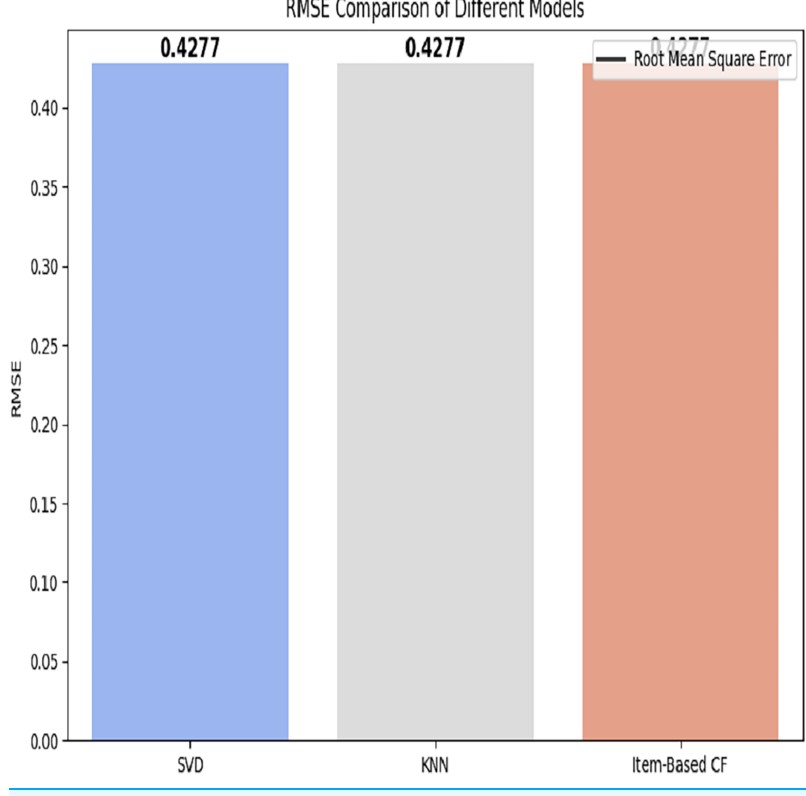

**Figure 9** **RMSE comparison of different models.**

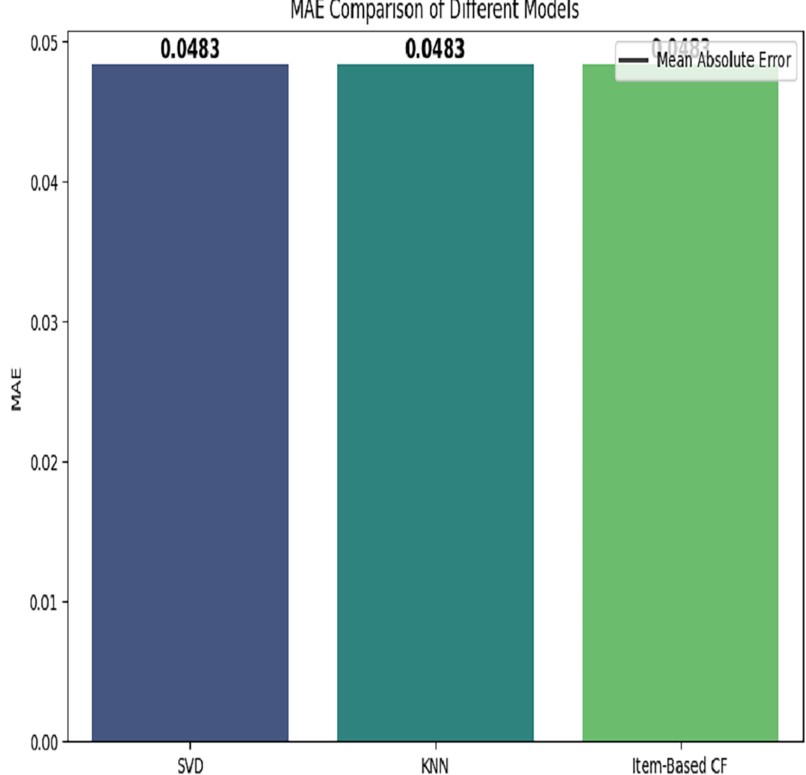

**Figure 10 MAE based comparison of different models.**

**Table 8 Insurance plan for category A.**

| Employee age | Emloyee gender | Marital status | Insurance benefit amount | Premium amount |
|---|---|---|---|---|
| 30.0 | Male | Married | 2,484,225 | 55,000 |
| 30.0 | Female | Married | 250,591 | 13,200 |

**Table 9 Insurance plan for category B.**

| Employee age | Emloyee gender | Marital status | Insurance benefit amount | Premium amount |
|---|---|---|---|---|
| 26 | Male | Married | 66,212 | 33,000 |
| 30 | Female | Married | 40,518 | 20,010 |

employees' data for the last 5 years. There are seven categories of insurance plans in the provided dataset. These categories are associated with the designation of employees. The most valuable employees are in category G. The most insurance coverage is for this category. We have generated customizeable insurance plans for each category, which adds flexibility for employees. The total amount for each category is determined from the historical medical records of each category. The generated customizable insurance plans are shown in Tables 8, 9, 10, 11, 12, 13 and 14.

**Table 10  Insurance plan for category C.**

| Employee age | Emloyee gender | Marital status | Total amount | |
|---|---|---|---|---|
| 28 | Female | Married | 2,460,744 | 17,208 |
| 29 | Male | Married | 18,693,312 | 58,784 |

**Table 11  Insurance plan for category D.**

| Age | Gender | Married status | Insurance benefit amount | Premium amount |
|---|---|---|---|---|
| 28 | Male | Married | 2,488,048 | 54,008 |
| 29 | Female | Married | 1,697,484 | 10,802 |

**Table 12  Insurance plan for category E.**

| Age | Gender | Relation status | Insurance benefit amount | Premium amount |
|---|---|---|---|---|
| 28 | Male | Married | 30,557,735 | 54,665 |
| 29 | Female | Married | 4,182,006 | 13,802.0 |

**Table 13  Insurance plan for category F.**

| Age | Gender | Marital status | Insurance benefit amount | Premium amount |
|---|---|---|---|---|
| 28 | Male | Married | 10,750,851 | 41,191.0 |
| 26 | Female | Single | 8,171,936 | 7,768.0 |

**Table 14  Insurance plan for category G.**

| Age | Gender | Marital status | Insurance benefit amount | Premium amount |
|---|---|---|---|---|
| 30.0 | 0.0 | 1.0 | 3,779,568 | 96,912.0 |
| 30.0 | 1.0 | 1.0 | 1,631,448 | 19,656.0 |

Table 8 depicts the insurance plan for the employees who belong to category A. As we have divided employees using clustering into two groups. Employees who are married and male need a higher insurance amount than female and unmarried employees. Enterprises can update it by adjusting the mean and variance values.

Similarly, in Tables 9, 10, 11, 12, 13 and 14, insurance plans for categories B, C, D, E, F and G are depicted.

Each category has two groups, and a different coverage amount is for each group within each category. Figure 11 depicts the percentile of each category. We can see from the plot that the probability distribution of each category is *Gaussian* distribution. The parameters for the Gaussian distribution are the mean and the variance. We have used these parameters to compute each category's total health coverage amount.

From Table 15, we can see the reviews for the Department of Cardiology. The root mean square is computed to evaluate the correctness of our proposed methodology. By using the set of predictions $\{\hat{y}_i\}$ and actual values $\{y_i\}$, the RMSE is computed.

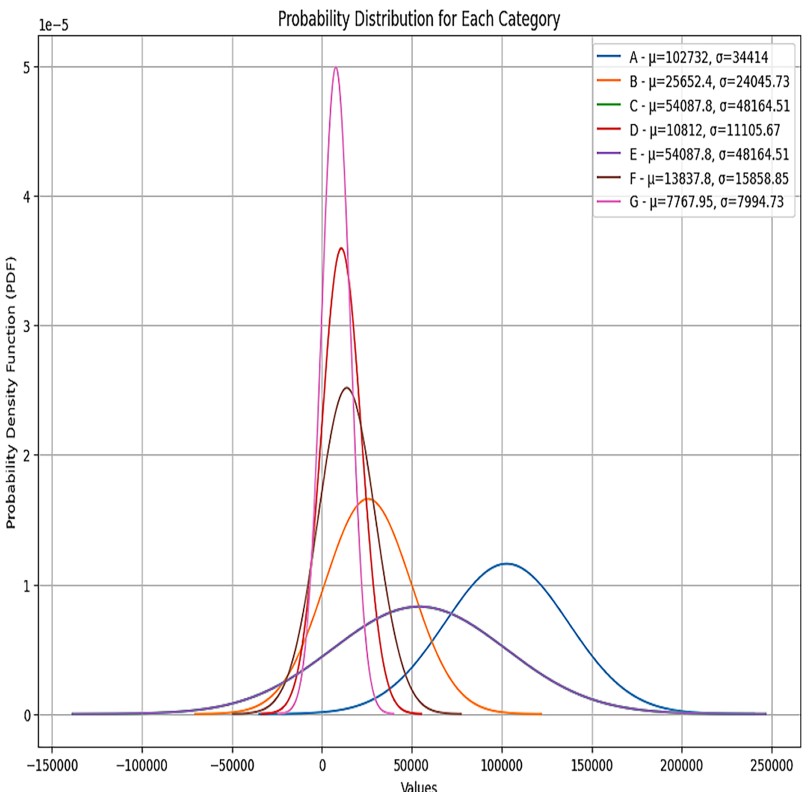

**Figure 11 Probability distributions for each category.**

**Table 15 Recommendations for Cardiology.**

| Serial | doctor_name | hospital_name | Department | Fee | Reviews | Similarity |
|---|---|---|---|---|---|---|
| 0 | H2D009 | H2 | Cardiology | 3,000 | 5.0 | 0.97 |
| 1 | H4D028 | H4 | Cardiology | 3,300 | 4.0 | 0.999983 |
| 2 | H2D005 | H2 | Cardiology | 3,000 | 3.0 | 0.999315 |
| 3 | H2D001 | H2 | Cardiology | 3,000 | 4.0 | 0.98 |
| 4 | H2D002 | H2 | Cardiology | 3,000 | 3.0 | 0.97 |
| 5 | H2D003 | H2 | Cardiology | 3,000 | 4.0 | 0.9999 |

Also, we compare the proposed recommender system with another doctor's recommender system, which is available on Marham.pk. Both platforms are designed to assist users in finding suitable doctors, but their approaches and features differ. The proposed recommender system implies multiple techniques to give both personalized and generalized recommendations. It analyzes the behaviour of similar users to provide accurate and relevant recommendations based on the good conduct of the doctors. In contrast, Marham.pk uses patient reviews to recommend doctors and generates generalized recommendations.

In terms of time and space complexity, both systems may have their trade-offs. The proposed system requires calculating similarity scores using cosine similarity, which can be

computationally intensive. However, by utilizing collaborative filtering, the system can provide more personalized recommendations, leading to a higher likelihood of a positive experience for users. Marham.pk, with its reliance on patient reviews, may have a less complex recommendation process. However, it may also have limitations in terms of providing personalized recommendations since it primarily relies on generalized information from patient reviews. The proposed Doctor Recommender System aims to provide accurate and relevant recommendations by leveraging collaborative filtering techniques. While it may involve some additional computational complexity, the personalized nature of the recommendations enhances the user experience. But in terms of performance, our recommender system gives much better recommendations by giving users a choice between generalized and personalized recommendations.

## Automated insurance claims reimbursement

The healthcare intelligent reimbursement insurance contract was successfully deployed on the blockchain utilizing the Remix IDE and the Metamask-injected provider environment. As a result, we developed a safe and transparent system for healthcare reimbursement that addresses a number of challenges prevalent in the current healthcare sector.

Firstly, the patient needs to be verified by the insurance company before the patient's claim is submitted, as shown in Algorithm 5; otherwise, the healthcare provider is not able to submit the claim on behalf of the patient. The claim submission involves two main fields as input: the claim ID and the amount of the treatment. Upon submitting the claim, the front-end interface would trigger a call to the "submitclaim" function of the smart contract using Web3, passing claim details to the Insurance Company.

Following the healthcare provider's submission of the claim, the insurance provider approves the claim as indicated in Algorithm 3. The "Approveclaim" function of the smart contract would be called when the insurance company approves the claim, supplying the claim ID as a parameter. If the claim is approved, the front end can display a success message that the claim has been approved.

The reimbursement process is automated, eliminating the need for intermediaries and reducing processing times and associated costs.

However, it is important to recognise the constraints and challenges encountered throughout the implementation. Blockchain networks may have problems when handling a high number of transactions at once due to their limited ability to scale. Additionally, users who are not familiar with blockchain technologies may encounter challenges while interacting with the Metamask browser extension.

We have carried out thorough evaluations using various criteria to assess the efficiency and functionality of the proposed Healthcare Reimbursement system. The following evaluation methods were used:

1. **Performance evaluation:** We measured the system's processing times for transactions such as claim submissions and claim acceptance or rejection. We compared these times with conventional reimbursement processes to evaluate the system's effectiveness and potential time savings.

2. **System security assessment:** The system's security was assessed by looking at the robustness of the blockchain infrastructure and smart contract implementation. We evaluated the system's resistance to frequent security risks such as tampering, unauthorised access, and data breaches. We also looked at the measures to safeguard patient data privacy.

In a nutshell, the proposed system uses SVD to address the sparsity in user-item interactions by reducing dimensionality. In KNN-based CF, historical ratings are used to find related doctor trends. Item-based CF assists with cold-start problems by predicting preferences based on item similarities. Content-based filtering using TF-IDF gives physicians names to generate suggestions based on textual similarity. The system's Prediction accuracy is efficiently measured using RMSE and MAE. For handling the sparse matrix, the missing value problem was avoided by setting fill_value = 0 in pivot_table. We have tackled the cold-start problem by implementing a hybrid strategy. We guarantee that new users and items receive pertinent recommendations by integrating collaborative filtering with demographic information (age, gender) and content-based elements (medical specializations, reviews).

By incorporating demographic features with collaborative filtering, the system improves by providing better similarity estimation. Users who have similar demographic information also have similar medical requirements; when there are sparse explicit ratings, even then, it is important to improve similarity computations. New users can get recommendations based on the demographic groups. As we know, medical recommendations are normally affected by age and gender; e.g., gynaecologists for women, contextually related recommendations are more useful. Our hybrid approach have not only reduced RMSE and MAE, when we include age and gender related information because they improve internal structure of user-item matrix, it has also increased the system's overall accuracy and relevance, which makes use of content-based filtering and demographic similarity to offset the absence of previous encounters.

The technique can scale across departments with different features since it incorporates flexible attribute handling and normalization to handle heterogeneous data. It uses tools like these to guarantee equity and prevent bias-sensitive attributes from being left out or modified in distance estimates. The financial values of the members of each cluster are balanced through dynamic centroid adjustments. Centroid-based matching reduces bias by guaranteeing homogeneous groups with comparable characteristics and financials. When used across many departments or datasets, these techniques together guarantee that the algorithm yields impartial, equitable clustering results.

The system uses distance-based clustering (*e.g.*, K-means) to handle data sparsity, and even for sparse groups, it can produce useful recommendations. The algorithm adjusts by using neighboring clusters or allocating sparse data points to the closest centroids if specific demographic groups (such as age or gender) have fewer users. This guarantees that significant recommendations are made to even underrepresented groups.

Furthermore, modifying the clustering features can tailor the system to other industries, like e-commerce and education. In e-commerce, it can leverage consumer behavior and

**Table 16 Insurance plan update results of proposed methodology.**

| Patient ID | Recommendation | Action taken | Updated plan ID | Txn hash (Blockchain) |
|------------|----------------|--------------|-----------------|------------------------|
| U001 | Upgrade to premium | Executed | P102 | 0 × 9a3…df5 |
| U002 | No change | Skipped | – | – |
| U003 | Include wellness plan | Executed | P107 | 0 × bc4…e18 |

**Table 17 Performance of the proposed system.**

| Module | Functionality/Outcome |
|--------|------------------------|
| Physician/Service recommendation | Similarity score up to 0.9 (personalized ranking) |
| Automation in claim processing (Smart Contracts) | 98.1% claims processed without Human Intervention |
| Data privacy | Role-based access + Off-chain storage + On-chain hash |
| Processing time | Reduced from 4–7 days to 24 h |
| Audit ease | Immutable logs with timestamps on-chain |

product preferences, while in education, it can group pupils according to academic achievement or learning styles. In both situations, the algorithm's capacity to provide tailored suggestions using clustering principles guarantees its adaptability to various domains.

By ensuring that demographic data is handled appropriately, we put controls in place to reduce bias. To lessen the emphasis on demographic-based suggestions, the system integrates several features, including patient reviews, treatment histories, and doctor specialities, rather than merely depending on demographic characteristics.

We work to uphold equity by making sure that the recommendations are supported by thorough information encompassing a variety of patient and physician profiles. This prohibits any certain demographic group from being unduly favoured by the system.

The system is intelligent as it generates customizeable insurance plans based on the history of each patient. Each time, a data-driven threshold is applied to the generated clusters within the methodology. A user consistently maintains his blood pressure goals, tracked by a wearable. The proposed recommender system recommends premium discounts or wellness plans. A smart contract updates the user policy terms and conditions on the chains. All entities, such as users, providers and regulators, can see this update.

Table 16 depicts how insurance plans are updated. We pledge to conduct routine audits of the model to make sure it does not discriminate against underrepresented groups or reinforce negative stereotypes. Feedback loops are used to adjust the system in response to actual results.

Blockchain technology (BT) is used for data integrity and automation, while ML is used for predictive analytics in the suggested smart and safe healthcare insurance system. By anticipating anomalous claims, suggesting qualified physicians and hospitals, and dynamically modifying insurance plans, machine learning algorithms like clustering, reinforcement learning similarity-based models improve decision-making. In the meantime, blockchain guarantees transparency, immutability, and stakeholder trust by keeping raw data off-chain and storing metadata and reference hashes on-chain. By automating claim

processing, smart contracts cut down on human intervention and delays. The model's potential for real-world scalability and security was demonstrated by a simulated evaluation using 10,000 medical records and 2,000 insurance claims, which showed 99% success in autonomous claim settlement and relevant physician recommendation. It can be seen from Table 17, that how we can evaluate performance of each of our module.

The evaluation process gave a thorough insight into the system's performance in resolving the challenges encountered during the medical insurance reimbursement procedure. These evaluation findings further contribute to the overall argument and conclusions presented in the article.

## CONCLUSION AND FUTURE WORK

In conclusion, developing recommendation systems and automating manual insurance claims reimbursements using blockchain technology represent significant strides in healthcare. By integrating factors such as age, location, gender, and ratings into the recommendation system, patients can now benefit from a personalised and tailored approach to finding suitable doctors. This improves the patient experience and enhances the overall quality of care. The novel concept of customizable insurance plans also enhances the system's applicability to the insurance industry. The proposed technique is validated on the 5-year historical employee data of the local hospital. Furthermore, leveraging blockchain technology to automate insurance claims reimbursement offers numerous advantages. Using blockchain ensures transparency, security, and efficiency, minimising errors and fraud. By streamlining this crucial aspect of healthcare administration, valuable time and resources can be saved, ultimately benefiting patients and healthcare providers.

In the future, this approach could incorporate patient reviews and doctor ratings from social media to acquire additional information about them, which would assist in enhancing the system's quality. The system could be further enhanced by including patient treatments and specific symptoms of a disease. The proposed approach may be linked to any current hospital management system, which may help patients choose the best physician when their health is at risk. Furthermore, building upon the existing doctor recommendation system, a potential avenue for further development would be incorporating natural language processing (NLP) capabilities. This would enable patients to enter their health queries or concerns in natural language, and based on these queries, relevant doctors would be recommended. The system must employ advanced NLP techniques, such as sentiment analysis, entity recognition, and semantic understanding, to accurately interpret and analyse the patient's queries. The system can provide more precise and tailored doctor recommendations by understanding the queries' context, intent, and underlying medical concepts.

## APPENDIX

This is our home page dashboard. It displays the location of hospitals along with their names. Also, the distance of the user to the hospitals is also displayed in kilometres. as depicted in Figs. A1, A2, A3.

Average ratings of hospitals given by user profiles are calculated in Fig. A2.

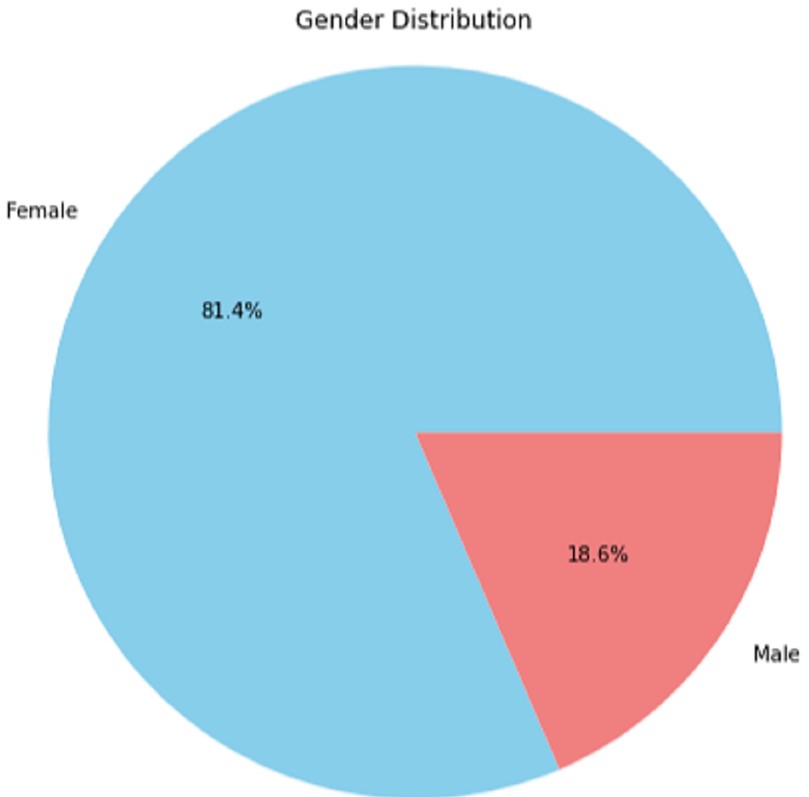

**Figure A1  Gender distribution graph.**

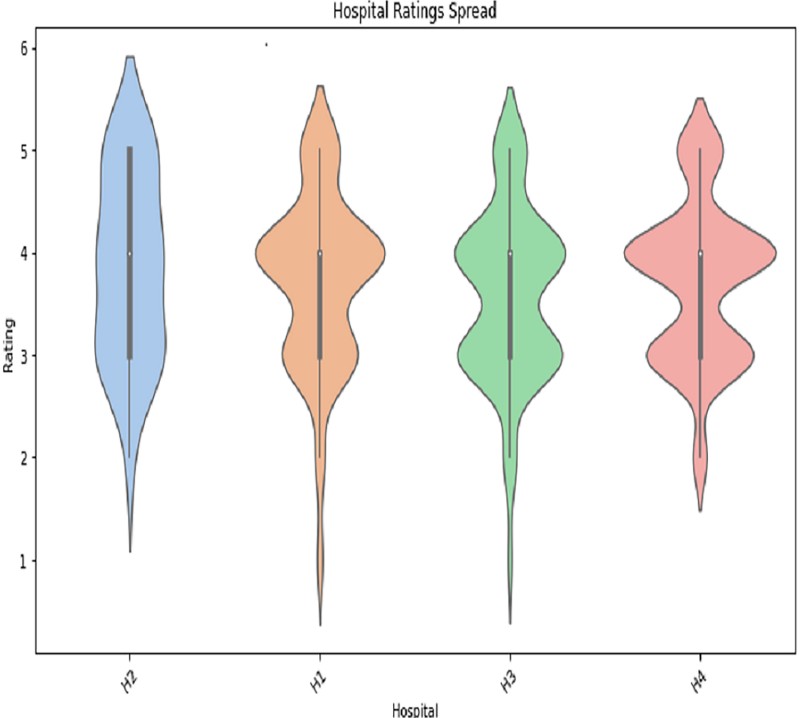

**Figure A2  Hospitals' ratings spread.**

## Find doctors by health concern

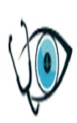 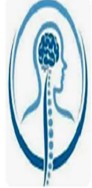 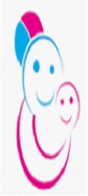 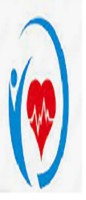 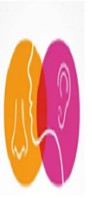 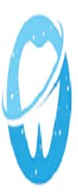 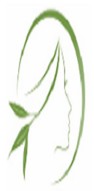 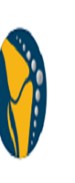 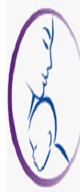

Eye Specialist · Neurologist · Child Specialist · Cardiologist · ENT Specialist · Dentist · Skin Specialist · Orthopedic Surgeon · Gynecologist

| Hospital Names | Location | Fee | Distance |
|---|---|---|---|
| COMBINED MILITARY HOSPITAL | H2JW+9V8, CMH Rd, Rawalpindi, Punjab 46000 | 1000-1500 | 4.31 km |
| SHIFA INTERNATIONAL HOSPITAL | Pitras Bukhari Road, Sector H-8/4, Islamabad | 3300 | 6.38 km |
| MILITARY HOSPITAL | Abid Majeed Rd, Rawalpindi. | 1000-1500 | 2.61 km |
| ARMED FORCES INSTITUTE OF CARDIOLOGY | Military Hospital Rd, Rawalpindi, Punjab 46000 | 1000-2000 | 2.61 km |

**Figure A3  Doctor's selection by department.** 

### Funding
This project is funded by Princess Nourah bint Abdulrahman University Researchers Supporting Project number (PNURSP2025R411), Princess Nourah bint Abdulrahman University, Riyadh, Saudi Arabia. The funders had a role in decision to publish. The funders had no role in study design, data collection and analysis, or preparation of the manuscript.

### Grant Disclosures
The following grant information was disclosed by the authors:
Princess Nourah bint Abdulrahman University, Riyadh, Saudi Arabia: PNURSP2025R411.

### Competing Interests
The authors declare that they have no competing interests.

### Author Contributions
- Irum Matloob conceived and designed the experiments, performed the experiments, analyzed the data, performed the computation work, prepared figures and/or tables, authored or reviewed drafts of the article, and approved the final draft.
- Shoab Khan conceived and designed the experiments, performed the experiments, authored or reviewed drafts of the article, and approved the final draft.

- Bushra Bashir analyzed the data, performed the computation work, authored or reviewed drafts of the article, and approved the final draft.
- Rukaiya Rukaiya performed the computation work, authored or reviewed drafts of the article, and approved the final draft.
- Javed Ali Khan performed the computation work, authored or reviewed drafts of the article, and approved the final draft.
- Hessa Alfraihi analyzed the data, authored or reviewed drafts of the article, and approved the final draft.

## Data Availability

Project files and the dataset used are available in the Supplemental Information.

The sources used to extract the data are:

Shifa hospital (https://patientportal.shifa.com.pk/book_appt).

Healthwire (https://healthwire.pk/doctors).

Marham (https://www.marham.pk/).

## Supplemental Information

Supplemental information for this article can be found online at http://dx.doi.org/10.7717/peerj-cs.2980#supplemental-information.

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
