# Peer review of "Data driven healthcare insurance system using machine learning and blockchain technologies"

_PeerJ Computer Science, doi:10.7717/peerj-cs.2980_

## Round 0.1 · original submission · Major Revisions

Dear authors,

You are advised to critically respond to all comments point by point when preparing an updated version of the manuscript and while preparing for the rebuttal letter. Please address all comments/suggestions provided by reviewers, considering that these should be added to the new version of the manuscript.

Kind regards,
PCoelho

·

Basic reporting

This manuscript presents an intelligent recommender system for healthcare and blockchain-based insurance management. While the topic is interesting, there are several areas that require detailed attention.

Firstly, the introduction should address the security vulnerabilities in current healthcare frameworks and explain how blockchain can mitigate these issues. This will help readers understand the importance of blockchain for healthcare.

The related work section is underdeveloped and should be improved. Avoid phrases like "This Paper," and review additional relevant research on blockchain in healthcare.

Experimental design

In Figure 2, please clarify the meaning of the different clusters presented.

Validity of the findings

Currently, the results section reads more like a student project report than a research study with meaningful findings. For instance, Figures 5, 6, and 7 merely display visualizations of existing data, which may not add substantial value to the experimental results. Additionally, the manuscript includes numerous screenshots of web pages that do not appear to be part of any experiments.

The experimental results should clearly focus on the performance and effectiveness of the proposed recommender system algorithm. It is also essential to conduct a comprehensive performance evaluation of the system.

Please elaborate on the necessity of blockchain for the insurance reimbursement process. Specifically, clarify what data you intend to store on the blockchain and whether a centralized database is also in use.

Include information on the consensus algorithm used in the blockchain component.

Performance comparisons with state-of-the-art methods should be included to demonstrate the value of the proposed system.

Reviewer 2 ·

Basic reporting

The proposed system, utilizing collaborative filtering, clustering, and blockchain technologies, aims to revolutionize healthcare by recommending doctors, generating personalized insurance plans, and preventing fraud. This innovative approach enhances security, transparency, and cost-effectiveness. While the research demonstrates promise, major revisions are necessary to enhance its quality and fully realize its potential within the healthcare industry.

Experimental design

How does the integration of blockchain technology ensure the security and transparency of insurance claims within the system, and what specific challenges related to interoperability, regulatory compliance, scalability, and data privacy arise when attempting to scale this solution across diverse healthcare providers and insurers operating in different regulatory environments?


The discussed approach integrates demographic profiling (age and gender) with collaborative filtering to overcome standard CF limitations such as sparsity, lack of contextual awareness, and cold-start issues. By narrowing user comparisons to demographically similar groups, the system aims to enhance recommendation accuracy and relevance. These enhancements prompt critical evaluations regarding their measurable impact on performance and usability. Discuss how does this approach improve upon standard collaborative filtering techniques? Does incorporating age and gender profiles provide measurable gains in accuracy or relevance?

The healthcare-specific recommender system utilizes demographic profiling, such as age and gender, to enhance the personalization of doctor recommendations. By considering these attributes, the system aims to tailor recommendations to the unique needs of individual users, improving relevance. However, this approach also brings forth potential ethical challenges and the risk of bias. I would like authors to discuss the unique challenges this algorithm address in healthcare-specific recommender systems. Specifically, how does it address ethical concerns, such as potential biases in doctor recommendations, and what safeguards are in place to ensure that the system promotes equitable access to care without reinforcing stereotypes or disproportionately favoring certain groups?

Validity of the findings

The need-based insurance plan generation algorithm uses K-means clustering to group users dynamically, aiming for personalization and cost optimization. In contrast, traditional rule-based approaches often rely on static, predefined rules that lack adaptability to user-specific needs. By leveraging clustering and centroid recalibration, the algorithm strives to refine grouping and better align insurance plans with individual requirements. This raises critical questions about the algorithm's practical advantages over conventional methods and its measurable benefits in real-world applications. How does this method perform compared to traditional rule-based insurance grouping or alternative clustering techniques? Are there benchmarks showing improved cost savings or satisfaction using this algorithm over conventional models?

What were the performance benchmarks for ensuring the system’s security, and how did blockchain improve data integrity compared to traditional methods? what were the key security measures implemented to protect sensitive data?

It is not very clear about the algorithm how it handles convergence when records consistently fall outside the centroid mean during iterations. Discuss the measures that are in place to ensure computational efficiency with large datasets (e.g., millions of records)

Please discuss the algorithm capability of handle heterogeneous data when scaling to multiple departments or attribute. Are there any mechanisms are in place to ensure fairness and avoid bias in recommendations?

Additional comments

How does the system handle sparsity in user interactions or demographic data? For instance, if there are few users of a specific age group or gender, does the system still produce meaningful recommendations? Can this system adapt to other domains, such as education or e-commerce, where personalization is critical?

---

## Round 0.2 · Major Revisions

Dear authors,

After the previous revision round, some adjustments still need to be made. As a result, I once more suggest that you thoroughly follow the instructions provided by the reviewers to answer their inquiries clearly.

You are advised to critically respond to all comments point by point when preparing a new version of the manuscript and while preparing for the rebuttal letter. All the updates should be included in the new version of the manuscript.

Kind regards,
PCoelho

·

Basic reporting

Revised sufficiently.

Experimental design

Good

Validity of the findings

Good

Additional comments

May be accepted

Reviewer 2 ·

Basic reporting

The authors have revised the paper, but I believe there are still many issues that need to be resolved.

1) Title of the research paper and The proposed system aims is mismatched.
2) RELATED WORK needs to be resided keeping into consideration Intelligent and Secure Healthcare Insurance System using ML and BT
6) Figure 6. Generated Recommendations, Figure 12, 13, …..is it a figure?? Check all figures and improve its quality, some figures are difficult to read and not explain well in the text. Figure 15 and 16 is not required in a research paper...
7) The captions of the figures are too brief and lack sufficient explanation to help readers fully understand the content and purpose of each figure.
8) The overall quality of the English language is average, with limited coherence and insufficient logical connections between sentences, which sometime hinders readability and understanding.

Experimental design

3) Figure 1. Modules of the Proposed System
The flowchart ends with "Module 3: Blockchain Processing," but it lacks detail on how this secures or enhances data from Modules 1 and 2. Without clear integration points (e.g., validating recommendations or storing historical data), it risks being an inefficient add-on. Suggestion, Define how blockchain interacts with earlier modules—e.g., securing recommendation data or immutably storing historical visit records. Add a feedback mechanism where blockchain-verified data updates Modules 1 and 2 in real-time, improving accuracy and trust.

4) Figure 3: System Architecture has some potential issues and I have given some suggestions for improvements.
The flowchart shows distinct layers (Presentation, Application, Data, Blockchain), but it’s unclear how data flows seamlessly between them. For example, the recommendation engine in the application layer relies on patient and doctor data, yet there’s no explicit feedback loop to update the data layer with new claims or preferences, potentially leading to outdated recommendations. Suggestion to improve it by adding a bidirectional data flow between the application layer’s recommendation engine and the data layer. For instance, after generating recommendations, the system could update patient and doctor data with new claims or preferences, ensuring the data layer remains current and improves future suggestions.

The blockchain layer includes smart contracts, transaction validation, and a distributed ledger, but it lacks detail on how it handles real-time data updates or scales with large volumes of patient and doctor data. Suggestion to improve it by Specifying a mechanism in the blockchain layer for real-time transaction validation, such as off-chain data processing for initial claims checks, with only finalized data committed to the distributed ledger. This would reduce bottlenecks while maintaining security.

The presentation layer includes user authentication, verification, claims approval, and user preference management. However, the process doesn’t appear to include multi-factor authentication or fraud detection, risking unauthorized access or approvals. Suggestion, Upgrade the presentation layer with multi-factor authentication (e.g., biometrics or one-time passwords) and add a fraud detection step during claims approval to prevent unauthorized submissions.


5) Check Aligorithm1 and other as all algorithms not scientifically written, for example,
Aligorithm1: Ambiguous Loop Syntax in Line 2, Incorrect Condition Operator in Line 3, Broken Loop and Return Logic in Line 4, Lack of Initialization and Variable Clarity

Validity of the findings

Add more results and discussion how the proposed model claimed to be Intelligent and Secure Healthcare Insurance System using ML and BT

---

## Round 0.3 · accepted · Accept

Dear authors, we are pleased to verify that you meet the reviewer's valuable feedback to improve your research.

Thank you for considering PeerJ Computer Science and submitting your work.

Kind regards
PCoelho

Reviewer 2 ·

Basic reporting

The authors have revised the paper as per the comments given.

Experimental design

The authors have revised the paper as per the comments given.

Validity of the findings

The authors have revised the paper as per the comments given.